# Prediction Risk and Estimation Risk of the Ridgeless Least Squares Estimator under General Assumptions on Regression Errors

**Sungyoon Lee**
Department of Computer Science
Hanyang University
sungyoonlee@hanyang.ac.kr

**Sokbae Lee**
Department of Economics
Columbia University
sl3841@columbia.edu

## Abstract

In recent years, there has been a significant growth in research focusing on minimum $\ell_2$ norm (ridgeless) interpolation least squares estimators. However, the majority of these analyses have been limited to an unrealistic regression error structure, assuming independent and identically distributed errors with zero mean and common variance. In this paper, we explore prediction risk as well as estimation risk under more general regression error assumptions, highlighting the benefits of overparameterization in a more realistic setting that allows for clustered or serial dependence. Notably, we establish that the estimation difficulties associated with the variance components of both risks can be summarized through the trace of the variance-covariance matrix of the regression errors. Our findings suggest that the benefits of overparameterization can be extended to time series, panel and grouped data.

## 1 Introduction

Recent years have witnessed a fast growing body of work that analyzes minimum $\ell_2$ norm (ridgeless) interpolation least squares estimators (see, e.g., Bartlett et al., 2020; Hastie et al., 2022; Tsigler & Bartlett, 2023, and references therein). Researchers in this field were inspired by the ability of deep neural networks to accurately predict noisy training data with perfect fits, a phenomenon known as "double descent" or "benign overfitting" (e.g., Belkin et al., 2018; 2019; 2020; Zou et al., 2021; Mei & Montanari, 2022, among many others). They discovered that to achieve this phenomenon, overparameterization is critical.

In the setting of linear regression, we have the training data $\{(x_i, y_i) \in \mathbb{R}^p \times \mathbb{R} : i = 1, \cdots, n\}$, where the outcome variable $y_i$ is generated from

$$y_i = x_i^\top \beta + \varepsilon_i, \ i = 1, \ldots, n,$$

$x_i$ is a vector of features (or regressors), $\beta$ is a vector of unknown parameters, and $\varepsilon_i$ is a regression error. Here, $n$ is the sample size of the training data and $p$ is the dimension of the parameter vector $\beta$.

In the literature, the main object for the theoretical analyses has been mainly on the out-of-sample prediction risk. That is, for the ridge or interpolation estimator $\hat{\beta}$, the literature has focused on

$$\mathbb{E}\Big[(x_0^\top \hat{\beta} - x_0^\top \beta)^2 \mid x_1, \ldots, x_n\Big],$$

where $x_0$ is a test observation that is identically distributed as $x_i$ but independent of the training data. For example, Dobriban & Wager (2018); Wu & Xu (2020); Richards et al. (2021); Hastie et al. (2022) analyzed the predictive risk of ridge(less) regression and obtained exact asymptotic expressions under the assumption that $p/n$ converges to some constant as both $p$ and $n$ go to infinity. Overall, they found the double descent behavior of the ridgeless least squares estimator in terms of the prediction risk. Bartlett et al. (2020); Kobak et al. (2020); Tsigler & Bartlett (2023) characterized the phenomenon of benign overfitting in a different setting.

To the best of our knowledge, a vast majority of the theoretical analyses have been confined to a simple data generating process, namely, the observations are independent and identically distributed (i.i.d.), and the regression errors have mean zero, have the common variance, and are independent of the feature vectors. That is,

$$(y_i, x_i^\top)^\top \sim \text{i.i.d. with } \mathbb{E}[\varepsilon_i] = 0, \mathbb{E}[\varepsilon_i^2] = \sigma^2 < \infty \text{ and } \varepsilon_i \text{ is independent of } x_i. \quad (1)$$

This assumption, although convenient, is likely to be unrealistic in various real-world examples. For instance, Liao et al. (2023) adopted high-dimensional linear models to examine the double descent phenomenon in economic forecasts. In their applications, the outcome variables include S&P firms' earnings, U.S. equity premium, U.S. unemployment rate, and countries' GDP growth rate. As in their applications, economic forecasts are associated with time series or panel data. As a result, it is improbable that (1) holds in these applications. As another example, Spiess et al. (2023) examined the performance of high-dimensional synthetic control estimators with many control units. The outcome variable in their application is the state-level smoking rates in the Abadie et al. (2010) dataset. Considering the geographical aspects of the U.S. states, it is unlikely that the regression errors underlying the synthetic control estimators adhere to (1). In short, it is desirable to go beyond the simple but unrealistic regression error assumption given in (1).

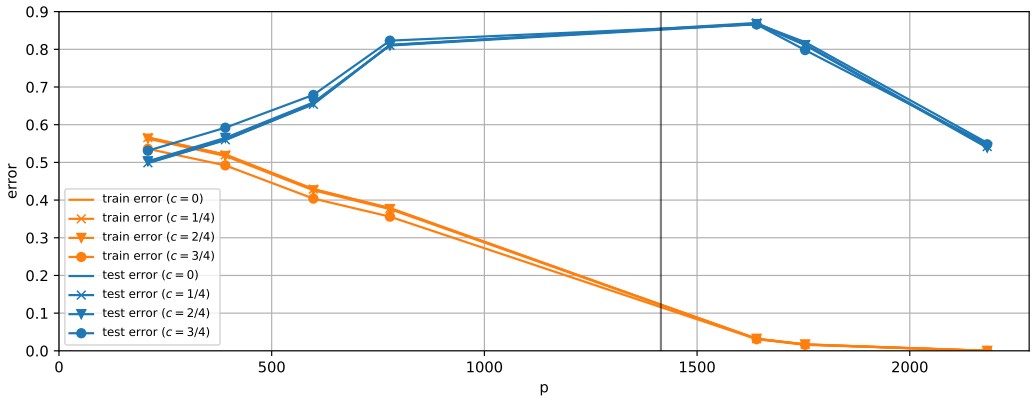

Figure 1: Comparison of in-sample and out-of-sample mean squared error (MSE) across various degrees of clustered noise. The vertical line indicates $p = n \ (= 1,415)$.

To further motivate, we start with our own real-data example from American Community Survey (ACS) 2018, extracted from IPUMS USA (Ruggles et al., 2024). The ACS is an ongoing annual survey by the US Census Bureau that provides key information about the US population. To have a relatively homogeneous population, the sample extract is restricted to white males residing in California with at least a bachelor's degree. We consider a demographic group defined by their age, the type of degree, and the field of degree. Then, we compute the average of log hourly wages for each age-degree-field group, treat each group average as the outcome variable, and predict group wages by various group-level regression models where the regressors are constructed using the indicator variables of age, degree, and field as well as their interactions. We consider 7 specifications ranging from 209 to 2,182 regressors. To understand the role of non-i.i.d. regressor errors, we add the artificial noise to the training sample. See Appendix A for details regarding how to generate the artificial noise. In the experiment, the constant $c$ varies such that $c = 0$ corresponds to no clustered dependence across observations but as a positive $c$ gets larger, the noise has a larger share of clustered errors but the variance of the overall regression errors remains the same regardless of the value of $c$. Figure 1 shows the in-sample (train) vs. out-of-sample (test) mean squared error (MSE) for various values of $c \in \{0, 0.25, 0.5, 0.75\}$. It can be seen that the experimental results are almost identical across different values of $c$ especially when $p > n$, suggesting that the double descent phenomenon might be universal for various degrees of clustered dependence, provided that the overall variance of the regression errors remains the same. It is our main goal to provide a firm foundation for this empirical phenomenon. To do so, we articulate the following research questions:

- How to analyze the out-of-sample prediction risk of the ridgeless least squares estimator under *general* assumptions on the regression errors?

- Why does *not* the prediction risk seem to be affected by the degrees of dependence across observations?

To delve into the prediction risk, suppose that $\Sigma := \mathbb{E}[x_0 x_0^\top]$ is finite and positive definite. Then,

$$\mathbb{E}\Big[(x_0^\top \hat{\beta} - x_0^\top \beta)^2 \mid x_1, \ldots, x_n\Big] = \mathbb{E}\Big[(\hat{\beta} - \beta)^\top \Sigma (\hat{\beta} - \beta) \mid x_1, \ldots, x_n\Big].$$

If $\Sigma = I$ (i.e., the case of isotropic features), where $I$ is the identity matrix, the mean squared error of the estimator defined by $\mathbb{E}[\|\hat{\beta} - \beta\|^2]$, where $\|\cdot\|$ is the usual Euclidean norm, is the same as the expectation of the prediction risk defined above. However, if $\Sigma \neq I$, the link between the two quantities is less intimate. One may regard the prediction risk as the $\Sigma$-weighted mean squared error of the estimator; whereas $\mathbb{E}[\|\hat{\beta} - \beta\|^2]$ can be viewed as an "unweighted" version, even if $\Sigma \neq I$. In other words, regardless of the variance-covariance structure of the feature vector, $\mathbb{E}[\|\hat{\beta} - \beta\|^2]$ treats each component of $\beta$ "equally." The mean squared error of the estimator is arguably one of the most standard criteria to evaluate the quality of the estimator in statistics. For instance, in the celebrated work by James & Stein (1961), the mean squared error criterion is used to show that the sample mean vector is not necessarily optimal even for standard normal vectors (so-called "Stein's paradox"). Many follow-up papers used the same criterion; e.g., Hansen (2016) compared the mean-squared error of ordinary least squares, James–Stein, and Lasso estimators in an underparameterized regime. Both $\Sigma$-weighted and unweighted versions of the mean squared error are interesting objects to study. For example, Dobriban & Wager (2018) called the former "predictive risk" and the latter "estimation risk" in high-dimensional linear models; Berthier et al. (2020) called the former "generalization error" and the latter "reconstruction error" in the context of stochastic gradient descent for the least squares problem using the noiseless linear model. In this paper, we analyze both weighted and unweighted mean squared errors of the ridgeless estimator under general assumptions on the data-generating processes, not to mention anisotropic features. Furthermore, our focus is on the finite-sample analysis, that is, both $p$ and $n$ are fixed but $p > n$.

Although most of the existing papers consider the simple setting as in (1), our work is not the first paper to consider more general regression errors in the overparameterized regime. Chinot et al. (2022); Chinot & Lerasle (2023) analyzed minimum norm interpolation estimators as well as regularized empirical risk minimizers in linear models without any conditions on the regression errors. Specifically, Chinot & Lerasle (2023) showed that, with high probability, without assumption on the regression errors, for the minimum norm interpolation estimator, $(\hat{\beta} - \beta)^\top \Sigma (\hat{\beta} - \beta)$ is bounded from above by $\left(\|\beta\|^2 \sum_{i \geq c \cdot n} \lambda_i(\Sigma) \vee \sum_{i=1}^n \varepsilon_i^2\right)/n$, where $c$ is an absolute constant and $\lambda_i(\Sigma)$ is the eigenvalues of $\Sigma$ in descending order. Chinot & Lerasle (2023) also obtained the bounds on the estimation error $(\hat{\beta} - \beta)^\top (\hat{\beta} - \beta)$. Our work is distinct and complements these papers in the sense that we allow for a general variance-covariance matrix of the regression errors. The main motivation of not making any assumptions on $\varepsilon_i$ in Chinot et al. (2022) and Chinot & Lerasle (2023) is to allow for potentially adversarial errors. We aim to allow for a general variance-covariance matrix of the regression errors to accommodate time series and clustered data, which are common in applications. See, e.g., Hansen (2022) for a textbook treatment (see Chapter 14 for time series and Section 4.21 for clustered data).

The main contribution of this paper is that we provide *exact finite-sample* characterization of the variance component of the prediction and estimation risks under the assumption that $X = [x_1, x_2, \cdots, x_n]^\top$ is *left-spherical* (e.g., $x_i$'s can be i.i.d. normal with mean zero but more general); $\varepsilon_i$'s *can be correlated and have non-identical variances*; and $\varepsilon_i$'s are independent of $x_i$'s. Specifically, the variance term can be factorized into a product between two terms: one term depends only on the *trace* of the variance-covariance matrix, say $\Omega$, of $\varepsilon_i$'s; the other term is solely determined by the distribution of $x_i$'s. Interestingly, we find that although $\Omega$ may contain non-zero off-diagonal elements, only the trace of $\Omega$ matters, as hinted by Figure 1, and further demonstrate our finding via numerical experiments. In addition, we obtain exact finite-sample expression for the bias terms when the regression coefficients follow the random-effects hypothesis (Dobriban & Wager, 2018). Our finite-sample findings offer a distinct viewpoint on the prediction and estimation risks, contrasting with the asymptotic inverse relationship (for optimally chosen ridge estimators) between the predictive and estimation risks uncovered by Dobriban & Wager (2018). Finally, we connect our findings to the existing results on the prediction risk (e.g., Hastie et al., 2022) by considering the asymptotic behavior of estimation risk. Remarkably, our findings stand in sharp contrast

to the well-established results in econometrics. In the latter, unlike in our framework, one of the key objectives is to estimate the variance-covariance matrix, denoted by $V_{\text{LS}}$, of the asymptotic distribution of the least squares estimators. In this context, the off-diagonal elements of $\Omega$ *do* affect $V_{\text{LS}}$, implying that any consistent estimator of $V_{\text{LS}}$ must account for these off-diagonal components.

One of the limitations of our theoretical analysis is that the design matrix $X$ is assumed to be left-spherical, although it is more general than i.i.d. normal with mean zero. We not only view this as a convenient assumption but also expect that our findings will hold at least approximately even if $X$ does not follow the left-spherical distribution. It is a topic for future research to formally investigate this conjecture.

## 2 THE FRAMEWORK UNDER GENERAL ASSUMPTIONS ON REGRESSION ERRORS

We first describe the minimum $\ell_2$ norm (ridgeless) interpolation least squares estimator in the over-parameterized case ($p > n$). Our goal is to understand the generalization ability of overparameterized models trained with gradient-based optimization (e.g., gradient descent) Gunasekar et al. (2017). Define

$$y := [y_1, y_2, \cdots, y_n]^\top \in \mathbb{R}^n,$$
$$\varepsilon := [\varepsilon_1, \varepsilon_2, \cdots, \varepsilon_n]^\top \in \mathbb{R}^n,$$
$$X^\top := [x_1, x_2, \cdots, x_n] \in \mathbb{R}^{p \times n},$$

so that $y = X\beta + \varepsilon$. The estimator we consider is

$$\hat{\beta} := \arg\min_{b \in \mathbb{R}^p}\{\|b\| : Xb = y\} = (X^\top X)^\dagger X^\top y = X^\dagger y,$$

where $A^\dagger$ denotes the Moore–Penrose inverse of a matrix $A$.

The main object of interest in this paper is the prediction and estimation risks of $\hat{\beta}$ under the data scenario such that the regression error $\varepsilon_i$ may *not* be i.i.d. Formally, we make the following assumptions.

**Assumption 2.1.** (i) $y = X\beta + \varepsilon$, where $\varepsilon$ is independent of $X$, and $\mathbb{E}[\varepsilon] = 0$. (ii) $\Omega := \mathbb{E}[\varepsilon\varepsilon^\top]$ is finite and positive definite (but not necessarily spherical).

We emphasize that Assumption 2.1 is more general than the standard assumption in the literature on benign overfitting that typically assumes that $\Omega \equiv \sigma^2 I$. Assumption 2.1 allows for non-identical variances across the elements of $\varepsilon$ because the diagonal elements of $\Omega$ can be different among each other. Furthermore, it allows for non-zero off-diagonal elements in $\Omega$. It is difficult to assume that the regression errors are independent among each other with time series or clustered data; thus, in these settings, it is important to allow for general $\Omega \neq \sigma^2 I$. Below we present a couple of such examples.

**Example 2.1** (Time Series - AR(1) Errors). Suppose that the regressor error follows an autoregressive process:

$$\varepsilon_i = \rho\varepsilon_{i-1} + \eta_i, \tag{2}$$

where $\rho \in (-1, 1)$ is an autoregressive parameter, $\eta_i$ is independent and identically distributed with mean zero and variance $\sigma^2(0 < \sigma^2 < \infty)$ and is independent of $X$. Then, the $(i, j)$ element of $\Omega$ is

$$\Omega_{ij} = \frac{\sigma^2}{1 - \rho^2}\rho^{|i-j|}.$$

Note that $\Omega_{ij} \neq 0$ as long as $\rho \neq 0$.

**Example 2.2** (Panel and Grouped Data - Clustered Errors). Suppose that regression errors are mutually independent across clusters but they can be arbitrarily correlated within the same cluster. For instance, students in the same school may affect each other and also have the same teachers; thus it would be difficult to assume independence across student test scores within the same school. However, it might be reasonable that student test scores are independent across different schools. For

example, assume that (i) if the regression error $\varepsilon_i$ belongs to cluster $g$, where $g = 1, \ldots, G$ and $G$ is the number of clusters, $\mathbb{E}[\varepsilon_i^2] = \sigma_g^2$ for some constant $\sigma_g^2 > 0$ that can vary over $g$; (ii) if the regression errors $\varepsilon_i$ and $\varepsilon_j$ $(i \neq j)$ belong to the same cluster $g$, $\mathbb{E}[\varepsilon_i \varepsilon_j] = \rho_g$ for some constant $\rho_g \neq 0$ that can be different across $g$; and (iii) if the regression errors $\varepsilon_i$ and $\varepsilon_j$ $(i \neq j)$ do not belong to the same cluster, $\mathbb{E}[\varepsilon_i \varepsilon_j] = 0$. Then, $\Omega$ is block diagonal with possibly non-identical blocks.

For vector $a$ and square matrix $A$, let $\|a\|_A^2 := a^\top A a$. Conditional on $X$ and given $A$, we define

$$\mathrm{Bias}_A(\hat{\beta} \mid X) := \|\mathbb{E}[\hat{\beta} \mid X] - \beta\|_A \quad \text{and} \quad \mathrm{Var}_A(\hat{\beta} \mid X) := \mathrm{Tr}(\mathrm{Cov}(\hat{\beta} \mid X)A),$$

and we write $\mathrm{Var} = \mathrm{Var}_I$ and $\mathrm{Bias} = \mathrm{Bias}_I$ for the sake of brevity in notation.

The mean squared prediction error for an unseen test observation $x_0$ with the positive definite covariance matrix $\Sigma := \mathbb{E}[x_0 x_0^\top]$ (assuming that $x_0$ is independent of the training data $X$) and the mean squared estimation error of $\hat{\beta}$ conditional on $X$ can be written as:

$$R_P(\hat{\beta} \mid X) := \mathbb{E}\big[(x_0^\top \hat{\beta} - x_0^\top \beta)^2 \mid X\big] = [\mathrm{Bias}_\Sigma(\hat{\beta} \mid X)]^2 + \mathrm{Var}_\Sigma(\hat{\beta} \mid X),$$

$$R_E(\hat{\beta} \mid X) := \mathbb{E}\big[\|\hat{\beta} - \beta\|^2 \mid X\big] = [\mathrm{Bias}(\hat{\beta} \mid X)]^2 + \mathrm{Var}(\hat{\beta} \mid X).$$

In what follows, we obtain exact finite-sample expressions for prediction and estimation risks:

$$R_P(\hat{\beta}) := \mathbb{E}_X[R_P(\hat{\beta} \mid X)] \quad \text{and} \quad R_E(\hat{\beta}) := \mathbb{E}_X[R_E(\hat{\beta} \mid X)].$$

We first analyze the variance terms for both risks and then study the bias terms.

## 3 THE VARIANCE COMPONENTS OF PREDICTION AND ESTIMATION RISKS

### 3.1 THE VARIANCE COMPONENT OF PREDICTION RISK

We rewrite the variance component of prediction risk as follows:

$$\mathrm{Var}_\Sigma(\hat{\beta} \mid X) = \mathrm{Tr}(\mathrm{Cov}(\hat{\beta} \mid X)\Sigma) = \mathrm{Tr}(X^\dagger \Omega X^{\dagger\top}\Sigma) = \|SX^\dagger T\|_F^2, \tag{3}$$

where positive definite symmetric matrices $S := \Sigma^{1/2}$ and $T := \Omega^{1/2}$ are the square root matrices of the positive definite matrices $\Sigma$ and $\Omega$, respectively. To compute the above Frobenius norm of the matrix $SX^\dagger T$, we need to compute the alignment of the right-singular vectors of $B := SX^\dagger \in \mathbb{R}^{p \times n}$ with the left-eigenvectors of $T \in \mathbb{R}^{n \times n}$. Here, $B$ is a random matrix while $T$ is fixed. Therefore, we need the distribution of the right-singular vectors of the random matrix $B$.

Perhaps surprisingly, to compute the *expected* variance $\mathbb{E}_X[\mathrm{Var}_\Sigma(\hat{\beta} \mid X)]$, it turns out that we do not need the distribution of the singular vectors if we make a minimal assumption (the *left-spherical symmetry* of $X$) which is weaker than the assumption that $\{x_i\}_{i=1}^n$ is i.i.d. normal with $\mathbb{E}[x_1] = 0$.

**Definition 3.1** (Left-Spherical Symmetry (Dawid, 1977; 1978; 1981; Gupta & Nagar, 1999)). *A random matrix $Z$ or its distribution is called to be left-spherical if $OZ$ and $Z$ have the same distribution $(OZ \overset{d}{=} Z)$ for any fixed orthogonal matrix $O \in O(n) := \{A \in \mathbb{R}^{n \times n} : AA^\top = A^\top A = I\}$.*

**Assumption 3.2.** *The design matrix $X$ is left-spherical.*

For the isotropic error case $(\Omega = I)$, we have $\mathbb{E}_X[\mathrm{Var}_\Sigma(\hat{\beta} \mid X)] = \mathbb{E}_X[\mathrm{Tr}((X^\top X)^\dagger \Sigma)]$ directly from (3) since $X^\dagger X^{\dagger\top} = (X^\top X)^\dagger$. Moreover, for the arbitrary error, the left-spherical symmetry of $X$ plays a critical role to *factor out* the same $\mathbb{E}_X[\mathrm{Tr}((X^\top X)^\dagger \Sigma)]$ and the trace of the variance-covariance matrix of the regression errors, $\mathrm{Tr}(\Omega)$, from the variance after the expectation over $X$.

**Lemma 3.3.** *For a subset $\mathcal{S} \subset \mathbb{R}^{m \times m}$ satisfying $C^{-1} \in \mathcal{S}$ for all $C \in \mathcal{S}$, if matrix-valued random variables $Z$ and $AZ$ have the same distribution measure $\mu_Z$ for any $A \in \mathcal{S}$, then we have*

$$\mathbb{E}_Z[f(Z)] = \mathbb{E}_Z[f(AZ)] = \mathbb{E}_Z[\mathbb{E}_{A' \sim \nu}[f(A'Z)]]$$

*for any function $f \in L^1(\mu_Z)$ and any probability density function $\nu$ on $\mathcal{S}$.*

**Theorem 3.4.** *Let Assumptions 2.1, and 3.2 hold. Then, we have*

$$\mathbb{E}_X[\mathrm{Var}_\Sigma(\hat{\beta} \mid X)] = \frac{1}{n} \mathrm{Tr}(\Omega)\mathbb{E}_X[\mathrm{Tr}((X^\top X)^\dagger \Sigma)].$$

*Sketch of Proof.* With $B = \Sigma^{1/2} X^\dagger$ and $T = \Omega^{1/2}$, we can rewrite the variance as follows:

$$\mathrm{Var}_\Sigma(\hat\beta \mid X) = \|BT\|_F^2 = \|UDV^\top U_T D_T V_T^\top\|_F^2 = \|DV^\top U_T D_T\|_F^2$$

from the singular value decompositions $B = UDV^\top$ and $T = U_T D_T V_T^\top$ with orthogonal matrices $U, V, U_T, V_T$, and diagonal matrices $D, D_T$. Then, we need to compute the alignment $V^\top U_T$ of the right-singular vectors of $B$ with the left-eigenvectors of $T$ because

$$\|DV^\top U_T D_T\|_F^2 = \lambda\left((X^\top X)^\dagger \Sigma\right)^\top \Gamma(X)\lambda(\Omega) = a(X)^\top \Gamma(X) b,$$

where $a(X) := \lambda\left((X^\top X)^\dagger \Sigma\right)$, $b := \lambda(\Omega)$, $v^{(i)} := V_{:i}$, $u^{(j)} := (U_T)_{:j}$, $\gamma_{ij} := \langle v^{(i)}, u^{(j)}\rangle^2 \geq 0$, $\Gamma(X) := (\gamma_{ij})_{i,j} \in \mathbb{R}^{n \times n}$ and $\lambda(A) \in \mathbb{R}^n$ is a vector where its elements are the eigenvalues of $A$.

Now, we want to compute the expected variance. To do so, from Lemma 3.3 with $\mathcal{S} = O(n)$ and the left-spherical symmetry of $X$, we can obtain

$$\mathbb{E}_X[a(X)^\top \Gamma(X) b] = \mathbb{E}_X\left[\mathbb{E}_{O\sim\nu}[a(OX)^\top \Gamma(OX) b]\right] = \mathbb{E}_X\left[a(X)^\top \mathbb{E}_{O\sim\nu}[\Gamma(OX)]b\right],$$

where $\nu$ is the unique uniform distribution (the Haar measure) over the orthogonal matrices $O(n)$.

Here, we can show that $\mathbb{E}_{O\sim\nu}[\Gamma(OX)] = \frac{1}{n}J$, where $J$ is the all-ones matrix with $J_{ij} = 1(i,j = 1, 2, \cdots, n)$. Therefore, we have the expected variance as follows:

$$\mathbb{E}_X[\mathrm{Var}_\Sigma(\hat\beta \mid X)] = \mathbb{E}_X\left[a(X)^\top \frac{1}{n} Jb\right] = \frac{1}{n}\sum_{i,j=1}^n \mathbb{E}_X[a_i(X)]b_j = \frac{1}{n}\mathbb{E}_X[\mathrm{Tr}((X^\top X)^\dagger \Sigma)]\,\mathrm{Tr}(\Omega).$$

$\square$

This recovers the previous results (e.g., Hastie et al. (2022)) when $\Omega = \sigma^2 I$, i.e., $\varepsilon_i$'s are i.i.d. with $\mathrm{Var}[\varepsilon_i] = \sigma^2$. The proofs of Lemma 3.3 and Theorem 3.4 are in the supplementary appendix.

## 3.2 THE VARIANCE COMPONENT OF ESTIMATION RISK

For the expected variance $\mathbb{E}_X[\mathrm{Var}(\hat\beta \mid X)]$ of the estimation risk, a similar argument still holds if plugging-in $B = X^\dagger$ instead of $B = \Sigma^{1/2} X^\dagger$.

**Theorem 3.5.** *Let Assumptions 2.1, and 3.2 hold. Then, we have*

$$\mathbb{E}_X[\mathrm{Var}(\hat\beta \mid X)] = \frac{1}{np}\mathrm{Tr}(\Omega)\mathbb{E}_X[\mathrm{Tr}(\Lambda^\dagger)],$$

*where $XX^\top / p = U\Lambda U^\top$ for some orthogonal matrix $U \in O(n)$.*

## 3.3 NUMERICAL EXPERIMENTS

In this section, we validate our theory with some numerical experiments of Examples 2.1 and 2.2, especially how the expected variance is related to the general covariance $\Omega$ of the regressor error $\varepsilon$. In the both examples, we sample $\{x_i\}_{i=1}^n$ from $\mathcal{N}(0, \Sigma)$ with a general feature covariance $\Sigma = U_\Sigma D_\Sigma U_\Sigma^\top$ for an orthogonal matrix $U_\Sigma \in O(p)$ and a diagonal matrix $D_\Sigma \succ 0$. In this setting, we have $\mathrm{rank}(XX^\top) = n$ and $\Lambda^\dagger = \Lambda^{-1}$ almost everywhere.

See Fig 6 in Appendix C for the experiments with large $n$ and $p$ (e.g., $n = 10\mathrm{k}$, $p = 150\mathrm{k}$).

**AR(1) Errors** As shown in Example 2.1, when the regressor error follows an autoregressive process in equation 2, we have $\Omega_{ij} = \sigma^2 \rho^{|i-j|}/(1 - \rho^2)$ and $\mathrm{Tr}(\Omega)/n = \sigma^2/(1 - \rho^2)$. Therefore, for pairs of $(\sigma^2, \rho^2)$ with the same $\mathrm{Tr}(\Omega)/n$, they are expected to yield the same variances of the prediction and estimation risk from Theorem 3.4 and 3.5 even though they have different off-diagonal elements in $\Omega$. To be specific, the pairs $(\sigma^2, \rho^2)$ on a line $\{(\sigma^2, \rho^2) : \sigma^2/\kappa^2 + \rho^2 = 1\}$ have the same $\mathrm{Tr}(\Omega)/n$ and the same expected variance which gets larger for the line with respect to a larger $\kappa^2$.

Figure 2 (left) shows the contour plots of $\mathbb{E}_X[\mathrm{Var}_\Sigma(\hat\beta \mid X)]$ and $\frac{1}{n}\mathrm{Tr}(\Omega)\mathbb{E}_X[\mathrm{Tr}((X^\top X)^\dagger \Sigma)]$ for different pairs of $(\sigma^2, \rho^2)$ in Example 2.1. They have different slopes $-\kappa^{-2}$ according to the value of $\kappa^2 = \mathrm{Tr}(\Omega)/n$. The right panel shows equivalent contour plots for estimation risk.

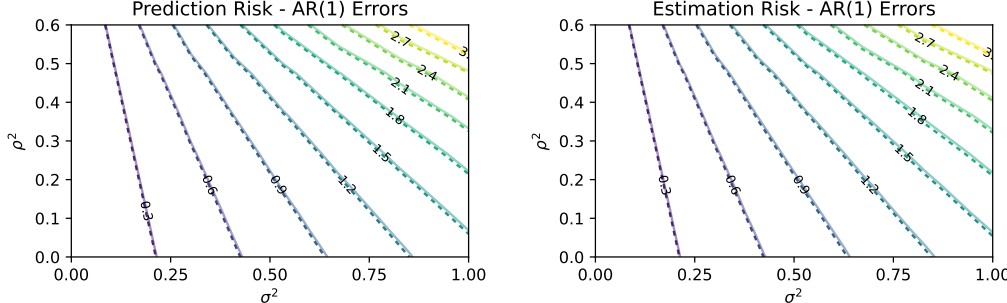

Figure 2: Our theory (dashed lines) matches the expected variances (solid lines) of the prediction (left) and estimation risks (right) in Example 2.1 (AR(1) Errors). Each point $(\sigma^2, \rho^2)$ represents a different noise covariance matrix $\Omega$, but with the same $\text{Tr}(\Omega)$ along each line $\{(\sigma^2, \rho^2) : \sigma^2/\kappa^2 + \rho^2 = 1\}$ for some $\kappa^2 > 0$, they have the same expected variance. We set $n = 500, p = 1000$, and evaluate on 100 samples of $X$ and 100 samples of $\varepsilon$ (for each realization of $X$) to approximate the expectations.

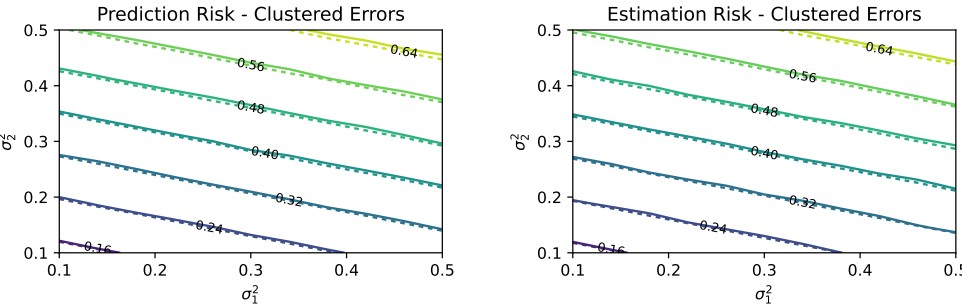

Figure 3: Our theory (dashed lines) matches the expected variances (solid lines) of the prediction (left) and estimation risks (right) in Example 2.2 (Clustered Errors). Each point $(\sigma_1^2, \sigma_2^2)$ represents a different noise covariance matrix $\Omega$, but with the same $\text{Tr}(\Omega)$ along each line $\{(\sigma_1^2, \sigma_2^2) : \frac{n_1}{n}\sigma_1^2 + \frac{n_2}{n}\sigma_2^2 = \kappa^2\}$ for some $\kappa^2 > 0$, they have the same expected variance. We set $G = 2, (n_1 = 50, n_2 = 150), n = 200, p = 400, \rho_1 = \rho_2 = 0.05$, and evaluate on 100 samples of $X$ and 100 samples of $\varepsilon$ (for each realization of $X$) to approximate the expectations.

**Clustered Errors** Now consider the block diagonal covariance matrix $\Omega = \text{diag}(\Omega_1, \Omega_2, \cdots, \Omega_G)$ in Example 2.2, where $\Omega_g$ is an $n_g \times n_g$ matrix with $(\Omega_g)_{ii} = \sigma_g^2$ and $(\Omega_g)_{ij} = \rho_g$ ($i \neq j$) for each $i, j = 1, 2, \cdots, n_g$ and $g = 1, 2, \cdots, G$. Let $n = \sum_{g=1}^{G} n_g$. We then have $\text{Tr}(\Omega)/n = \sum_{g=1}^{G} \text{Tr}(\Omega_g)/n = \sum_{g=1}^{G}(n_g/n)\sigma_g^2$. Therefore, given a partition $\{n_g\}_{g=1}^{G}$ of the $n$ observations, the covariance matrices $\Omega$ with different $\{\sigma_g^2\}_{g=1}^{G}$ have the same $\text{Tr}(\Omega)/n$ if $(\sigma_1^2, \sigma_2^2, \cdots, \sigma_G^2) \in \mathbb{R}^G$ are on the same hyperplane $\frac{n_1}{n}\sigma_1^2 + \frac{n_2}{n}\sigma_2^2 + \cdots + \frac{n_G}{n}\sigma_G^2 = \kappa^2$ for some $\kappa^2 > 0$.

Figure 3 (left) shows the contour plots of $\mathbb{E}_X[\text{Var}_\Sigma(\hat{\beta} \mid X)]$ and $\frac{1}{n}\text{Tr}(\Omega)\mathbb{E}_X[\text{Tr}((X^\top X)^\dagger \Sigma)]$ for different pairs of $(\sigma_1^2, \sigma_2^2)$ for a simple two-clusters example ($G = 2$) of Example 2.2 with $(n_1, n_2) = (5, 15)$. Here, we use a fixed value of $\rho_1 = \rho_2 = 0.05$, but the results are the same regardless of their values, as shown in the Appendix. Unlike Example 2.1, the hyperplanes are orthogonal to $v = [n_1, n_2]^\top$ regardless of the value of $\kappa^2 = \text{Tr}(\Omega)/n$. Again, the right panel shows equivalent contour plots for estimation risk.

## 4  THE BIAS COMPONENTS OF PREDICTION AND ESTIMATION RISKS

Our main contribution is to allow for general assumptions on the regression errors, and thus the bias parts remain the same as they do not change with respect to the regression errors. For completeness, in this section, we briefly summarize the results on the bias components. First, we make the following assumption for a constant rank deficiency of $X^\top X$ which holds, for example, each $x_i$ has a positive definite covariance matrix and is independent of each other.

**Assumption 4.1.** $\mathrm{rank}(X) = n$ almost everywhere.

### 4.1  THE BIAS COMPONENT OF PREDICTION RISK

The bias term of prediction risk can be expressed as follows:

$$[\mathrm{Bias}_\Sigma(\hat\beta \mid X)]^2 = (S\beta)^\top \lim_{\lambda \searrow 0} \lambda^2 (S^{-1}\hat\Sigma S + \lambda I)^{-2} S\beta, \tag{4}$$

where $\hat\Sigma := X^\top X/n$. Now, in order to obtain an exact closed form solution, we make the following assumption:

**Assumption 4.2.** $\mathbb{E}_\beta[S\beta(S\beta)^\top] = r_\Sigma^2 I/p$, where $r_\Sigma^2 := \mathbb{E}_\beta[\|\beta\|_\Sigma^2] < \infty$ and $\beta$ is independent of $X$.

A similar assumption (see Assumption 4.4) has been shown to be useful to obtain closed-form expressions in the literature (e.g., Dobriban & Wager, 2018; Richards et al., 2021; Li et al., 2021; Chen et al., 2023).

Under this assumption, since $[\mathrm{Bias}_\Sigma(\hat\beta \mid X)]^2 = \mathrm{Tr}[S\beta(S\beta)^\top \lim_{\lambda \searrow 0} \lambda^2 (S^{-1}\hat\Sigma S + \lambda I)^{-2}]$ from (4), we have the expected bias (conditional on $X$) as follows:

$$\mathbb{E}_\beta[\mathrm{Bias}_\Sigma(\hat\beta \mid X)^2 \mid X] = \frac{r_\Sigma^2}{p} \lim_{\lambda \searrow 0} \sum_{i=1}^p \frac{\lambda^2}{(\tilde s_i + \lambda)^2} = \frac{r_\Sigma^2}{p} |\{i \in [p] : \tilde s_i = 0\}| = r_\Sigma^2 \frac{p-n}{p},$$

where $\tilde s_i$ are the eigenvalues of $S^{-1}\hat\Sigma S \in \mathbb{R}^{p \times p}$ and $\mathrm{rank}(S^{-1}\hat\Sigma S) = \mathrm{rank}(X) = n$ almost everywhere under Assumption 4.1. This bias is independent of the distribution of $X$ or the spectral density of $S^{-1}\hat\Sigma S$, but only depending on the rank deficiency of the realization of $X$.

Finally, the prediction risk $R_P(\hat\beta)$ can be summarized as follows:

**Corollary 4.3.** *Let Assumptions 2.1, 3.2, 4.1, and 4.2 hold. Then, we have*

$$R_P(\hat\beta) = r_\Sigma^2 \left(1 - \frac{n}{p}\right) + \frac{\mathrm{Tr}(\Omega)}{n} \mathbb{E}_X \left[\mathrm{Tr}((X^\top X)^\dagger \Sigma)\right].$$

### 4.2  THE BIAS COMPONENT OF ESTIMATION RISK

For the bias component of estimation risk, we can obtain a similar result with 4.1 as follows:

$$[\mathrm{Bias}(\hat\beta \mid X)]^2 = \beta^\top (I - \hat\Sigma^\dagger \hat\Sigma)\beta = \lim_{\lambda \searrow 0} \beta^\top \lambda(\hat\Sigma + \lambda I)^{-1}\beta.$$

**Assumption 4.4.** $\mathbb{E}_\beta[\beta\beta^\top] = r^2 I/p$, where $r^2 := \mathbb{E}_\beta[\|\beta\|^2] < \infty$ and $\beta$ is independent of $X$.

Under Assumption 4.4, we have the expected bias (conditional on $X$) as follows:

$$\mathbb{E}_\beta[\mathrm{Bias}(\hat\beta \mid X)^2 \mid X] = \frac{r^2}{p} \lim_{\lambda \searrow 0} \sum_{i=1}^p \frac{\lambda}{s_i + \lambda} = \frac{r^2}{p} |\{i \in [p] : s_i = 0\}| = r^2 \frac{p-n}{p}, \tag{5}$$

where $s_i$ are the eigenvalues of $\hat\Sigma \in \mathbb{R}^{p \times p}$ and $\mathrm{rank}(\hat\Sigma) = \mathrm{rank}(X) = n$ under Assumption 4.1.

Thanks to Theorem 3.5 and (5), we obtain the following corollary for estimation risk.

**Corollary 4.5.** *Let Assumptions 2.1, 3.2, 4.1, and 4.4 hold. Then, we have*

$$R_E(\hat\beta) = r^2 \left(1 - \frac{n}{p}\right) + \frac{\mathrm{Tr}(\Omega)}{n} \mathbb{E}_X \left[\int \frac{1}{s} dF^{XX^\top/n}(s)\right],$$

*where $F^A(s) := \frac{1}{n} \sum_{i=1}^n 1\{\lambda_i(A) \le s\}$ is the empirical spectral distribution of a matrix $A$ and $\lambda_1(A), \lambda_2(A), \cdots, \lambda_n(A)$ are the eigenvalues of $A$.*

The proof of Corollary 4.5 is in the Appendix.

### 4.2.1 ASYMPTOTIC ANALYSIS OF ESTIMATION RISK

To study the asymptotic behavior of estimation risk, we follow the previous approaches (Dobriban & Wager, 2018; Hastie et al., 2022). First, we define the Stieltjes transform as follows:

**Definition 4.6.** The Stieltjes transform $s_F(z)$ of a df $F$ is defined as:

$$s_F(z) := \int \frac{1}{x - z} dF(x), \text{ for } z \in \mathbb{C} \setminus \text{supp}(F).$$

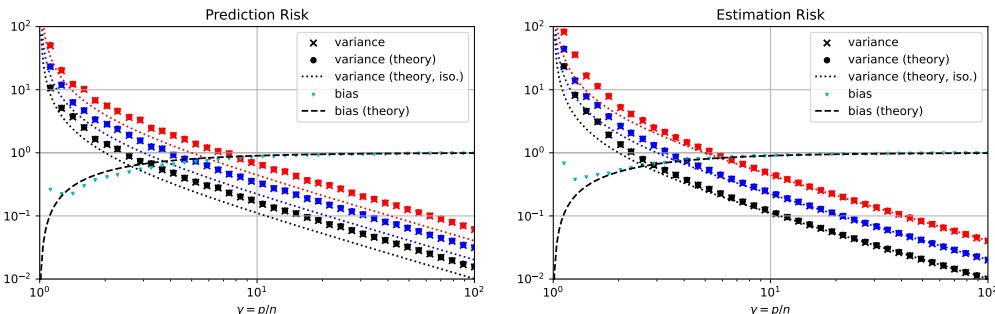

Figure 4: The "descent curve" in the overparameterization regime for prediction risk (left) and estimation risk (right). We test $\Omega$'s with $\text{Tr}(\Omega)/n = 1, 2, 4$ in black, blue, red, respectively. For the anisotropic feature, the expected variance ($\times$) and its theoretical expression ($\bullet$) are $\Theta\left(\frac{\text{Tr}(\Omega)/n}{\gamma - 1}\right)$ and larger than that in the high-dimensional asymptotics for the isotropic $\Sigma = I$. For the isotropic $\Sigma = I$, the variance terms (dotted) and the bias terms (dashed) in the high-dimensional asymptotics are $\frac{1}{\gamma - 1} \lim_{n \to \infty} \frac{\text{Tr}(\Omega)}{n}$ and $r^2\left(1 - \frac{1}{\gamma}\right)$, respectively.

We are now ready to investigate the asymptotic behavior of the mean squared estimation error with the following theorem:

**Theorem 4.7.** *(Silverstein & Bai, 1995, Theorem 1.1) Suppose that the rows $\{x_i\}_{i=1}^n$ in $X$ are i.i.d. centered random vectors with $\mathbb{E}[x_1 x_1^\top] = \Sigma$ and that the empirical spectral distribution $F^\Sigma(s) = \frac{1}{p}\sum_{i=1}^p 1\{\tau_i \leq s\}$ of $\Sigma$ converges almost surely to a probability distribution function $H$ as $p \to \infty$. When $p/n \to \gamma > 0$ as $n, p \to \infty$, then a.s., $F^{XX^\top/n}$ converges vaguely to a df $F$ and the limit $s^* := \lim_{z \searrow 0} s_F(z)$ of its Stieltjes transform $s_F$ is the unique solution to the equation:*

$$1 - \frac{1}{\gamma} = \int \frac{1}{1 + \tau s^*} dH(\tau). \tag{6}$$

This theorem is a direct consequence of Theorem 1.1 in Silverstein & Bai (1995). Then, from Corollary 4.5, we can write the limit of estimation risk as follows:

**Corollary 4.8.** *Let Assumptions 2.1, 3.2, 4.1, and 4.4 hold. Then, under the same assumption as Theorem 4.7, as $n, p \to \infty$ and $p/n \to \gamma$, where $1 < \gamma < \infty$ is a constant, we have*

$$R_E(\hat{\beta}) = \mathbb{E}\big[\|\hat{\beta} - \beta\|^2\big] \to r^2\left(1 - \frac{1}{\gamma}\right) + s^* \lim_{n \to \infty} \frac{\text{Tr}(\Omega)}{n}.$$

Here, the limit $s^*$ of the Stieltjes transform $s_F$ is highly connected with the shape of the spectral distribution of $\Sigma$. For example, in the case of isotropic features ($\Sigma = I$), i.e., $dH(\tau) = \delta(\tau - 1)d\tau$, we have $s^*_{\text{iso}} = (\gamma - 1)^{-1}$ from $1 - \frac{1}{\gamma} = \frac{1}{1 + s^*_{\text{iso}}}$. In addition, if $\Omega = \sigma^2 I$, then the limit of the mean squared error is exactly the same as the expression for $\gamma > 1$ in equation (10) of Hastie et al. (2022, Theorem 1). This is because prediction risk is the same as estimation risk when $\Sigma = I$.

*Remark* 4.9. Generally, if the support of $H$ is bounded within $[c_H, C_H] \subset \mathbb{R}$ for some positive constants $0 < c_H \leq C_H < \infty$, then we can observe the double descent phenomenon in the overparameterization regime with $\lim_{\gamma \searrow 1} s^* = \infty$ and $\lim_{\gamma \to \infty} s^* = 0$ with $s^* = \Theta\left(\frac{1}{\gamma-1}\right)$ from the following inequalities:

$$C_H^{-1} \frac{1}{\gamma - 1} \leq s^* \leq c_H^{-1} \frac{1}{\gamma - 1}. \tag{7}$$

In fact, a tighter lower bound is available:

$$s^* \geq \mu_H^{-1}(\gamma - 1)^{-1}, \tag{8}$$

where $\mu_H := \mathbb{E}_{\tau \sim H}[\tau]$, i.e., the mean of distribution $H$. The proofs of (7) and (8) are given in the supplementary appendix.

We conclude this paper by plotting the "descent curve" in the overparameterization regime in Figure 4. On one hand, the expected variance ($\times$) perfectly matches its theoretical counterpart ($\bullet$) and goes to zero as $\gamma$ gets large. On the other hand, the bias term is bounded even if $\gamma \to \infty$. The Appendix contains the experimental details for all the figures.

## 5 CONCLUSION

We present an analysis of the prediction and estimation risks of the minimum $\ell_2$ norm (ridgeless) interpolation least squares estimator under more general regression error assumptions, highlighting the benefits of overparameterization in a more realistic setting. This allows for clustered or serial dependence, which enables us to extend our results to time series, panel and grouped data. Notably, we provide an important understanding that the estimation difficulties associated with the variance components of both risks can be factorized into a product between two terms: one term depends only on the trace of the variance-covariance matrix of $\varepsilon_i$'s; the other term is solely determined by the distribution of $x_i$'s. Although $\Omega$ may contain non-zero off-diagonal elements, the off-diagonal correlation $\mathbb{E}[\varepsilon_i \varepsilon_j]$ do not play any role but only the trace of $\Omega$ matters. It would be a promising orthogonal research direction to explore the risks under a general feature setting such as kernel regression or random feature models, along with general error assumptions.

### ACKNOWLEDGMENTS

This work was partly supported by Institute of Information & communications Technology Planning & Evaluation (IITP) grants (RS-2020-II201373, Artificial Intelligence Graduate School Program (Hanyang University); RS-2023-002206284, Artificial intelligence for prediction of structure-based protein interaction reflecting physicochemical principles) and the National Research Foundation of Korea (NRF) grants (RS-2023-00244896, Implicit bias of optimization algorithms for robust generalization of deep learning; the BK21 FOUR (Fostering Outstanding Universities for Research) project; NRF-2024S1A5C3A02043653, Socio-Technological Solutions for Bridging the AI Divide: A Blockchain and Federated Learning-Based AI Training Data Platform) funded by the Korean government (MSIT).

### ETHICS STATEMENT

This paper presents work whose goal is to advance the field of Machine Learning. There are many potential societal consequences of our work, none which we feel must be specifically highlighted here.

### REPRODUCIBILITY STATEMENT

Detailed explanations to draw Fig 1-4 are included in Appendix A and B. We also attach our code to facilitate the reproduction of our experiments.

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

## A   DETAILS FOR DRAWING FIGURE 1

To draw Figure 1, we use a sample extract from American Community Survey (ACS) 2018.  To have a relatively homogeneous population, the sample extract is restricted to white males residing in California with at least a bachelor's degree. We consider a demographic group defined by their age in years (between 25 and 70), the type of degree (bachelor's, master's, professional, and doctoral), and the field of degree (172 unique values). Then, we compute the average of log hourly wages for each age-degree-field group (all together 7,073 unique groups in the sample).  We treat each group average as the outcome variable (say, $y_{a,d,f}$) and predict group wages by various group-level regression models where the regressors are constructed using the indicator variables of age, degree, and field as well as their interactions: that is,

$$y_{a,d,f} = x_{a,d,f}^\top \beta + \varepsilon_{a,d,f}.$$

For the regressors $x_{a,d,f}$, we consider 7 specifications ranging from 209 to 2,183 regressors:

- Spec. 1 ($p = 209$): dummy variables for age (say, $x_a$) + dummy variables for the type of degree (say, $x_d$) + dummy variables for the field of degree (say, $x_f$),
- Spec. 2 ($p = 391$): Spec. 1 + all interactions between $x_d$ and $x_a$,
- Spec. 3 ($p = 598$): Spec. 1 + all interactions between $x_d$ and $x_f$,
- Spec. 4 ($p = 778$): Spec. 1 + all interactions between $x_d$ and $x_a$ + all interactions between $x_d$ and $x_f$,
- Spec. 5 ($p = 1640$): Spec. 1 + all interactions between $x_d$ and $x_a$ + all interactions between $x_a$ and $x_f$,
- Spec. 6 ($p = 1754$): Spec. 1 + all interactions between $x_d$ and $x_f$ + all interactions between $x_a$ and $x_f$,
- Spec. 7 ($p = 2182$): Spec. 1 + all three-way interactions among $x_a$, $x_d$ and $x_f$.

Here, the dummy variable are constructed using one-hot encoding. We randomly split the sample into the train and test samples with a ratio of $1 : 4$. The resulting sample sizes are 1,415 and 5,658, respectively. To understand the role of non-i.i.d. regressor errors, we add the artificial noise to the training sample: that is, we compute the ridgeless least squares estimator using the training sample of $(\tilde{y}_{a,d,f}, x_{a,d,f}^\top)^\top$, where $\tilde{y}_{a,d,f} = y_{a,d,f} + u_{a,d,f}$. Here, the artificial noise $u_{a,d,f}$ has the form

$$u_{a,d,f} \equiv \frac{(1-c)e_{a,d,f} + c \cdot e_f}{\sqrt{(1-c)^2 + c^2}},$$

where $e_{a,d,f} \sim N(0, \sigma^2)$, independently across age ($a$), degree ($d$) and field ($f$); $e_f$ is the average of another independent $N(0, \sigma^2)$ variable within $f$ (hence, $e_f$ is identical for each value of $f$) and thus the source of clustered errors; and $c \in \{0, 0.25, 0.5, 0.75\}$ is a constant that will be varied across the experiment. As $c$ gets larger, the noise has a larger share of clustered errors but the variance of the overall regression errors ($u_{a,d,f}$) remains the same: in other words, $\mathrm{var}(u_{a,d,f}) = \sigma^2$ for each value of $c$. Figure 1 was generated with $\sigma = 0.5$ by generating the artificial noise only once.

# B    DETAILS FOR DRAWING FIGURES 2, 3, AND 4

To draw Figures 2, 3, and 4, we sample $\{x_i\}_{i=1}^n$ from $\mathcal{N}(0, \Sigma)$ with $\Sigma = U_\Sigma D_\Sigma U_\Sigma^\top$ where $U_\Sigma$ is an orthogonal matrix random variable, drawn from the uniform (Haar) distribution on $O(p)$, and $D_\Sigma$ is a diagonal matrix with its elements $d_i = |z_i|/\sum_{i=1}^p |z_i|$ being sampled with $z_i \sim \mathcal{N}(0, 1)$ for each $i = 1, 2, \cdots, p$. With this general anisotropic $\Sigma$, the term $\mathbb{E}_X[\mathrm{Tr}(\Lambda^{-1})]/p$ is somewhat larger than $\mu_H^{-1} s_{\mathrm{iso}}^* = (\gamma - 1)^{-1}$ which is 1 in Figures 2 and 3 since $\mu_H = 1$ and $\gamma = 2$. For example, in Figure 2, when $\sigma^2 = 1$, $\rho^2 = 0$, we have $\mathrm{Tr}(\Omega)/n = 1$ but $\mathrm{Tr}(\Omega)\mathbb{E}_X[\mathrm{Tr}(\Lambda^{-1})]/(np) > 1$.

In Figure 4, we fix $n = 50$ and use $p = n\gamma$ for $\gamma \in [1, 100]$.

To compute the expectations of $\mathbb{E}_X[\mathrm{Var}(\hat{\beta}|X)]$ and $\mathbb{E}_X[\mathrm{Tr}(\Lambda^{-1})]$ over $X$, we sample $N_X$ samples of $X$'s, $X_1, X_2, \cdots, X_{N_X}$. Moreover, to compute the expectation over $\varepsilon$ in $\mathrm{Var}(\hat{\beta}|X_i) \equiv \mathrm{Tr}\left(\mathbb{E}_\varepsilon[\hat{\beta}\hat{\beta}^\top] - \mathbb{E}_\varepsilon[\hat{\beta}]\mathbb{E}_\varepsilon[\hat{\beta}]^\top\right)$, we sample $N_\varepsilon$ samples of $\varepsilon$'s, $\varepsilon_1, \varepsilon_2, \cdots, \varepsilon_{N_\varepsilon}$ for each realization $X_i$. To be specific,

$$\mathbb{E}_X[\mathrm{Var}(\hat{\beta}|X)] \approx \frac{1}{N_X} \sum_{i=1}^{N_X} \mathrm{Var}(\hat{\beta}|X_i) \approx \frac{1}{N_X} \sum_{i=1}^{N_X} \mathrm{Tr}\left(\frac{1}{N_\varepsilon} \sum_{j=1}^{N_\varepsilon} \hat{\beta}_{i,j}\hat{\beta}_{i,j}^\top - \frac{1}{N_\varepsilon} \sum_{j=1}^{N_\varepsilon} \hat{\beta}_{i,j} \frac{1}{N_\varepsilon} \sum_{j=1}^{N_\varepsilon} \hat{\beta}_{i,j}^\top\right)$$

$$\frac{1}{p}\mathbb{E}_X[\mathrm{Tr}(\Lambda^{-1})] \approx \frac{1}{N_X} \sum_{i=1}^{N_X} \mathrm{Tr}((X_i X_i^\top)^{-1}) = \frac{1}{N_X} \sum_{i=1}^{N_X} \sum_{k=1}^n \frac{1}{\lambda_k(X_i X_i^\top)},$$

where $\hat{\beta}_{i,j} = \arg\min_\beta\{\|b\| : X_i b - y_{i,j} = 0\}$, $y_{i,j} = X_i \beta + \varepsilon_j$, and $\lambda_k(X_i X_i^\top)$ is the $k$-th eigenvalue of $X_i X_i^\top$. We can do similarly for the variance part of the prediction risk.

Figure 5 shows an additional experimental result.

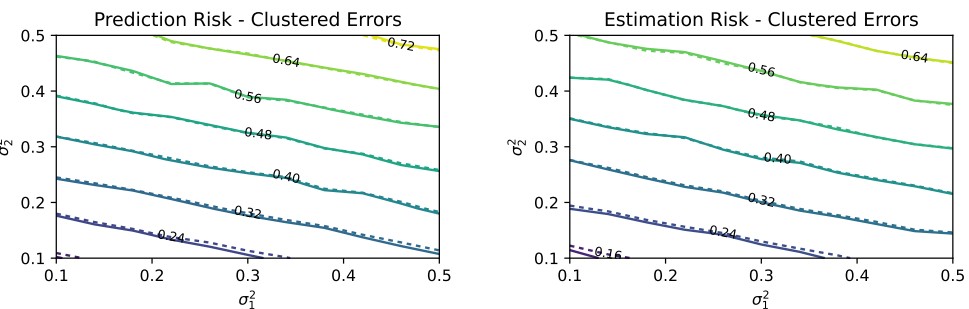

Figure 5:   We use a similar setting as Figure 3, except uniformly sample each $\rho_i$ from $[0, 0.05]$ for each experiment with the pairs $(\sigma_1^2, \sigma_1^2)$. As expected, the off-diagonal elements $\rho_i$ of $\Omega$ do not affect the expected variances.

# C EXPERIMENTS WITH LARGE $n$ AND $p$

We conduct the extra experiments with larger $n$ and $p$ (e.g., $n = 10$k and $p = 150$k).

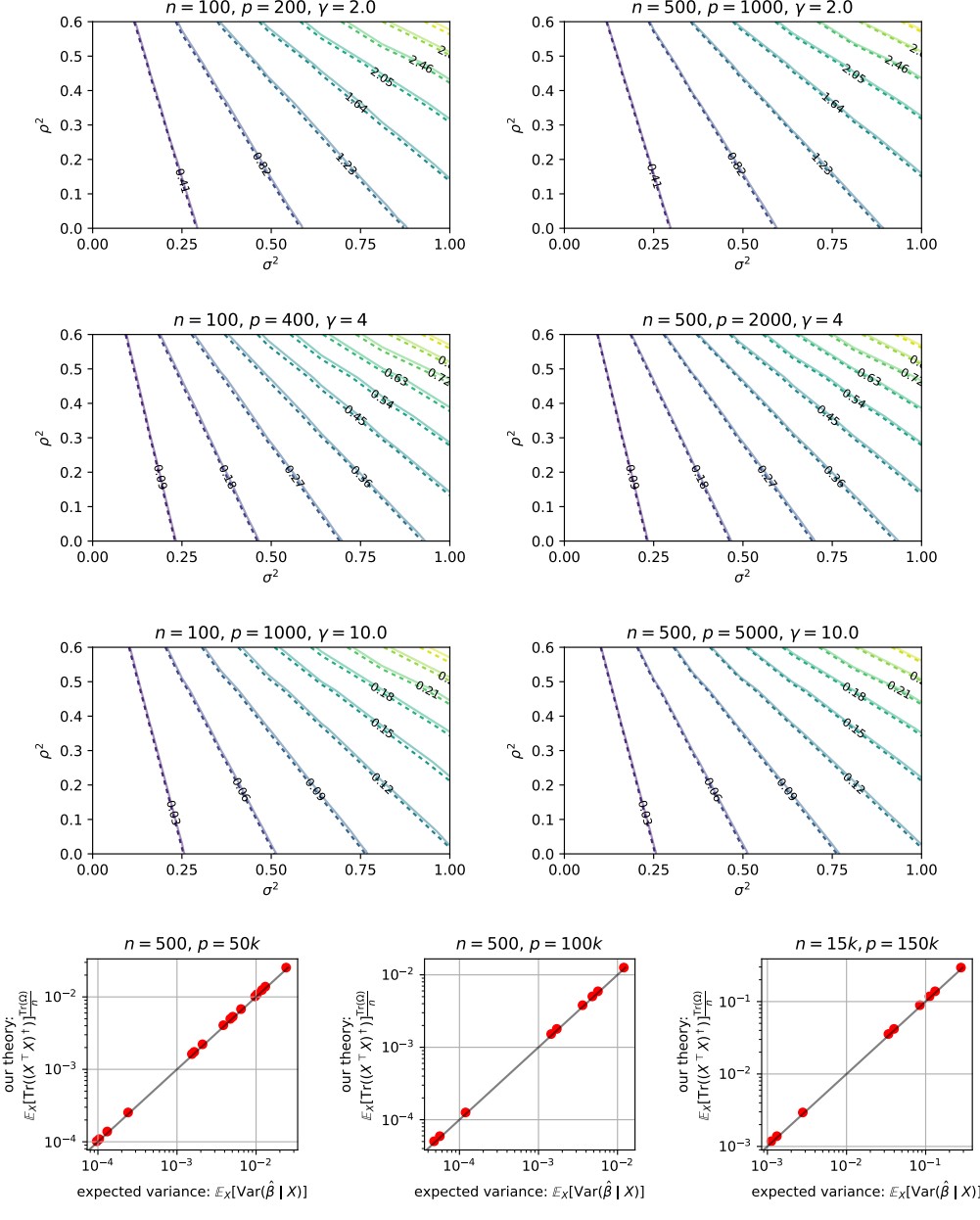

Figure 6: **[Top Three Rows]** Estimation Risk in Example 2.1 (AR(1) Errors) with a wide range of $(n, p)$ pairs. See caption of Fig 2 (Right) for more details. Panels in each row have the same $\gamma = 2, 4, 10$ (Top, Middle, Bottom). With the same $\gamma$, it shows almost identical results for each row. **[Last Row]** Large scale validation with $p = 50000 = 50$k (Left), $p = 100$k (Middle), and $p = 150$k (Rigth). cf. $x_i \in \mathbb{R}^p$ (CIFAR-10 $p \approx 3.1$k and ImageNet $p \approx 150$k).

## D    PROOFS OMITTED IN THE MAIN TEXT

*Proof of Lemma 3.3.* For a given $A \in \mathcal{S}$, since $A^{-1} \in \mathcal{S}$, we have $Z \stackrel{d}{=} A^{-1} Z := \tilde{Z}$ and

$$\mathbb{E}_Z[f(Z)] = \mathbb{E}_{A^{-1}Z}[f(Z)] = \mathbb{E}_{\tilde{Z}}[f(A\tilde{Z})] = \mathbb{E}_Z[f(AZ)].$$

This naturally leads to

$$\mathbb{E}_Z[\mathbb{E}_{A'\sim\nu}[f(A'Z)]] = \mathbb{E}_{A'\sim\nu}[\mathbb{E}_Z[f(A'Z)]] = \mathbb{E}_{A'\sim\nu}[\mathbb{E}_Z[f(Z)]] = \mathbb{E}_Z[f(Z)]$$

where the first equality comes from Fubini's theorem and the integrability of $f$.    $\square$

*Proof of Theorem 3.4.* Since $\hat{\beta} = X^\dagger y$, we have $\mathrm{Cov}(\hat{\beta} \mid X) = X^\dagger \mathrm{Cov}(y \mid X) X^{\dagger\top} = X^\dagger \Omega X^{\dagger\top}$, which leads to the following expression for the variance component of prediction risk:

$$\mathrm{Var}_\Sigma(\hat{\beta} \mid X) = \mathrm{Tr}(\mathrm{Cov}(\hat{\beta} \mid X)\Sigma) = \mathrm{Tr}(X^\dagger \Omega X^{\dagger\top}\Sigma) = \|SX^\dagger T\|_F^2 = \|BT\|_F^2,$$

where $S = \Sigma^{1/2}, T = \Omega^{1/2}$, and $B = SX^\dagger$. Using the singular value decomposition (SVD) of $B$ and $T$, respectively, we can rewrite this as follows:

$$\|BT\|_F^2 = \|UDV^\top U_T D_T V_T^\top\|_F^2 = \|DV^\top U_T D_T\|_F^2,$$

where $B = UDV^\top$ and $T = U_T D_T V_T^\top$ with orthogonal matrices $U, V, U_T, V_T$, and diagonal matrices $D, D_T$. Now we need to compute the alignment $V^\top U_T$ of the right-singular vectors of $B$ with the left-eigenvectors of $T$.

$$\begin{aligned}
\|DV^\top U_T D_T\|_F^2 &= \sum_{i,j=1}^n \left( D_{ii} \sum_{k=1}^n V_{ik}^\top (U_T)_{kj} (D_T)_{jj} \right)^2 \\
&= \sum_{i,j=1}^n \lambda_i(B)^2 \lambda_j(T)^2 \gamma_{ij} \\
&= \sum_{i,j=1}^n \lambda_i\left((X^\top X)^\dagger \Sigma\right) \lambda_j(\Omega) \gamma_{ij} \\
&= \underbrace{\lambda\left((X^\top X)^\dagger \Sigma\right)^\top}_{1 \times n} \underbrace{\Gamma(X)}_{n \times n} \underbrace{\lambda(\Omega)}_{n \times 1},
\end{aligned}$$

where $\gamma_{ij} := \langle V_{:i}, (U_T)_{:j} \rangle^2 \geq 0$, $\Gamma(X) := (\gamma_{ij})_{i,j} \in \mathbb{R}^{n \times n}$ and $\lambda(A) \in \mathbb{R}^n$ is a vector with its element $\lambda_i(A)$ as the $i$-th largest eigenvalue of $A$.

Therefore, we can rewrite the variance as $\mathrm{Var}_\Sigma(\hat{\beta} \mid X) = a(X)^\top \Gamma(X) b$ with

$$a(X) := \lambda\left((X^\top X)^\dagger \Sigma\right) \in \mathbb{R}^n,$$
$$b := \lambda(\Omega) \in \mathbb{R}^n,$$
$$\Gamma(X)_{ij} = \gamma_{ij} = \langle v^{(i)}, u^{(j)} \rangle^2,$$

where $v^{(i)} := V_{:i}$ and $u^{(j)} := (U_T)_{:j}$. Note that the alignment matrix $\Gamma(X)$ is a doubly stochastic matrix since $\sum_j \gamma_{ij} = \sum_i \gamma_{ij} = 1$ and $0 \leq \gamma_{ij} \leq 1$.

Now, we want to compute the expected variance. To do so, from Lemma 3.3 with $\mathcal{S} = O(n)$, we can obtain

$$\mathbb{E}_X[a(X)^\top \Gamma(X) b] = \mathbb{E}_X\left[\mathbb{E}_{O\sim\nu}[a(OX)^\top \Gamma(OX) b]\right] = \mathbb{E}_X\left[a(X)^\top \mathbb{E}_{O\sim\nu}[\Gamma(OX)] b\right],$$

where $\nu$ is the unique uniform distribution (the Haar measure) over the orthogonal matrices $O(n)$. For an orthogonal matrix $O \in O(n)$, we have

$$\Gamma(OX)_{ij} = \langle Ov^{(i)}, u^{(j)} \rangle^2 = (v^{(i)\top} O^\top u^{(j)})^2,$$

since $S(OX)^\dagger = SX^\dagger O^\top = BO^\top = UD(OV)^\top$. Here, $(OX)^\dagger = X^\dagger O^\top$ follows from the orthogonality of $O \in O(n)$. Since the Haar measure is invariant under the matrix multiplication in $O(n)$, if we take the expectation over the Haar measure, then we have

$$\bar{\Gamma}(X)_{ij} := \mathbb{E}_{O\sim\nu}[\Gamma(OX)_{ij}] = \mathbb{E}_{O\sim\nu}[(v^{(i)\top} O^\top u^{(j)})^2] = \mathbb{E}_{O\sim\nu}[(v^{(i)\top} O^\top O^{(j)\top} u^{(j)})^2].$$

Here, for a given $j$, we can choose a matrix $O^{(j)} \in O(n)$ such that its first column is $u^{(j)}$ and $O^{(j)\top} u^{(j)} = e_1$, then $\bar{\Gamma}(X)_{ij}$ is independent of $j$ (say $\bar{\Gamma}(X)_{ij} = \alpha_i$). Since $\Gamma(X)$ is doubly stochastic, so is $\bar{\Gamma}(X)$ and we have $\sum_{j=1}^n \bar{\Gamma}(X)_{ij} = n\alpha_i = 1$ which yields $\bar{\Gamma}(X)_{ij} = \alpha_i = 1/n$, regardless of the distribution of $V$; thus, $\bar{\Gamma}(X) = \frac{1}{n} J$, where $J_{ij} = 1(i, j = 1, 2, \cdots, n)$.

Therefore, we have the expected variance as follows:

$$\mathbb{E}_X[\mathrm{Var}_\Sigma(\hat{\beta} \mid X)] = \mathbb{E}_X[a(X)^\top \frac{1}{n} Jb] = \frac{1}{n} \sum_{i,j=1}^n \mathbb{E}_X[a_i(X)]b_j = \frac{1}{n} \mathbb{E}_X[\mathrm{Tr}((X^\top X)^\dagger \Sigma)] \, \mathrm{Tr}(\Omega).$$

$\square$

*Proof of Corollary 4.5.* Note that

$$\begin{aligned}
\mathbb{E}_X[\mathrm{Var}(\hat{\beta}|X)] &= \frac{\mathrm{Tr}(\Omega)}{p} \mathbb{E}_X\left[ \frac{1}{n} \sum_i \frac{1}{\lambda_i} \right] \\
&= \frac{\mathrm{Tr}(\Omega)}{p} \mathbb{E}_X\left[ \int \frac{1}{s} dF^{XX^\top/p}(s) \right] \\
&= \frac{\mathrm{Tr}(\Omega)}{n} \mathbb{E}_X\left[ \int \frac{1}{s} dF^{XX^\top/n}(s) \right].
\end{aligned}$$

Then, the desired result follows directly from (5). $\square$

*Proof of (4).* The bias term of the prediction risk can be expressed as follows:

$$\begin{aligned}
[\mathrm{Bias}_\Sigma(\hat{\beta} \mid X)]^2 &= \|\mathbb{E}[\hat{\beta} \mid X] - \beta\|_\Sigma^2 \\
&= \|(\hat{\Sigma}^\dagger \hat{\Sigma} - I)\beta\|_\Sigma^2 \\
&= \beta^\top (I - \hat{\Sigma}^\dagger \hat{\Sigma}) \Sigma (I - \hat{\Sigma}^\dagger \hat{\Sigma}) \beta \\
&= \beta^\top \lim_{\lambda \searrow 0} \lambda(\hat{\Sigma} + \lambda I)^{-1} \Sigma \lim_{\lambda \searrow 0} \lambda(\hat{\Sigma} + \lambda I)^{-1} \beta \\
&= (S\beta)^\top \lim_{\lambda \searrow 0} \lambda^2 (S^{-1} \hat{\Sigma} S + \lambda I)^{-2} S\beta,
\end{aligned}$$

where $\hat{\Sigma} = X^\top X/n$. Here, the fourth equality comes from the equation

$$\begin{aligned}
I - \hat{\Sigma}^\dagger \hat{\Sigma} &= \lim_{\lambda \searrow 0} I - (\hat{\Sigma} + \lambda I)^{-1} \hat{\Sigma} \\
&= \lim_{\lambda \searrow 0} I - (\hat{\Sigma} + \lambda I)^{-1} (\hat{\Sigma} + \lambda I - \lambda I) \\
&= \lim_{\lambda \searrow 0} \lambda(\hat{\Sigma} + \lambda I)^{-1}.
\end{aligned}$$

$\square$

*Proof of (7).* The RHS of (6) is bounded above by $\int \frac{1}{1+c_H s^*} dH(\tau) = \frac{1}{1+c_H s^*}$, and thus $1 - \frac{1}{\gamma} \leq \frac{1}{1+c_H s^*}$, which yields $s^* \leq c_H^{-1} \frac{1}{\gamma-1}$. We can similarly prove the other inequality in (7) with a lower bound $\frac{1}{1+C_H s^*}$ on the RHS of (6). $\square$

*Proof of (8).* To further explore the inequalities (7), we rewrite (6) from Theorem 4.7 as follows:

$$1 - \frac{1}{\gamma} = \mathbb{E}_{\tau \sim H}[g(\tau; s^*)], \quad \text{where } g(t; s) := \frac{1}{1+ts} \text{ for } t, s > 0.$$

Here, since $g(t; s)$ is convex with respect to $t > 0$ for a given $s > 0$, by Jensen's inequality, we then have

$$\mathbb{E}_{\tau \sim H}[g(\tau; \mu_H^{-1} s_{\text{iso}}^*)] \geq g\left(\mu_H; \mu_H^{-1} s_{\text{iso}}^*\right) = g(1; s_{\text{iso}}^*) = 1 - \gamma^{-1}$$

where $\mu_H = \mathbb{E}_{\tau \sim H}[\tau]$. Therefore, the limit Stieltjes transform $s^*$ in the anisotropic case should be larger than $\mu_H^{-1} s_{\text{iso}}^*$ of the isotropic case to satisfy $\mathbb{E}_{\tau \sim H}[g(\tau; s^*)] = 1 - \gamma^{-1}$ since $g(t; s)$ is a decreasing function with respect to $s \geq 0$ when $t > 0$. This leads to a tighter lower bound $s^* \geq \mu_H^{-1} s_{\text{iso}}^* = \mu_H^{-1} (\gamma - 1)^{-1}$ than (7) because $\mu_H \leq C_H$. $\square$

# E  RIDGE REGRESSION

We can easily extend Theorem 3.4 to the ridge regression case with the ridge estimator $\hat{\beta}_\lambda :=$ $\arg\min_{b \in \mathbb{R}^p}\{\|Xb - y\|^2 + \lambda\|b\|^2\}$, i.e., $\hat{\beta}_\lambda = (X^\top X + \lambda I)^{-1}X^\top y$.

**Theorem E.1.** *Let Assumptions 2.1, and 3.2 hold. Then, we have*

$$\mathbb{E}_X[\mathrm{Var}_\Sigma(\hat{\beta}_\lambda \mid X)] = \frac{1}{n}\mathrm{Tr}(\Omega)\mathbb{E}_X[\mathrm{Tr}((X^\top X + \lambda I)^{-1}X^\top X(X^\top X + \lambda I)^{-1}\Sigma)].$$

*Sketch of Proof.* The ridge estimator is $\hat{\beta}_\lambda = P_\lambda y$, where $P_\lambda = (X^\top X + \lambda I)^{-1}X^\top$. With $B_\lambda = \Sigma^{1/2}P_\lambda$ and $T = \Omega^{1/2}$, we can rewrite the variance as follows:

$$\mathrm{Var}_\Sigma(\hat{\beta}_\lambda \mid X) = \|B_\lambda T\|_F^2 = \|UD_\lambda V^\top U_T D_T V_T^\top\|_F^2 = \|D_\lambda V^\top U_T D_T\|_F^2$$

from the singular value decompositions $B_\lambda = UD_\lambda V^\top$ and $T = U_T D_T V_T^\top$ with orthogonal matrices $U, V, U_T, V_T$, and diagonal matrices $D_\lambda, D_T$. Note that $U, V$ do not depend on $\lambda$. Then, we need to compute the alignment $V^\top U_T$ of the right-singular vectors of $B_\lambda$ with the left-eigenvectors of $T$ because

$$\|D_\lambda V^\top U_T D_T\|_F^2 = \lambda\left(P_\lambda P_\lambda^\top \Sigma\right)^\top \Gamma(X)\lambda(\Omega) = a(X;\lambda)^\top\Gamma(X)b,$$

where $a(X;\lambda) := \lambda\left(P_\lambda P_\lambda^\top\Sigma\right) \in \mathbb{R}^n, b := \lambda(\Omega) \in \mathbb{R}^n, v^{(i)} := V_{:i}, u^{(j)} := (U_T)_{:j}, \gamma_{ij} := \langle v^{(i)}, u^{(j)}\rangle^2 \geq 0, \Gamma(X) := (\gamma_{ij})_{i,j} \in \mathbb{R}^{n \times n}$ and $\lambda(A) \in \mathbb{R}^n$ is a vector where its elements are the eigenvalues of $A$.

Now, we want to compute the expected variance. To do so, from Lemma 3.3 with $\mathcal{S} = O(n)$ and the left-spherical symmetry of $X$, we can obtain

$$\mathbb{E}_X[a(X;\lambda)^\top\Gamma(X)b] = \mathbb{E}_X\left[\mathbb{E}_{O \sim \nu}[a(OX;\lambda)^\top\Gamma(OX)b]\right] = \mathbb{E}_X\left[a(X;\lambda)^\top\mathbb{E}_{O \sim \nu}[\Gamma(OX)]b\right],$$

where $\nu$ is the unique uniform distribution (the Haar measure) over the orthogonal matrices $O(n)$.

Here, we can show that $\mathbb{E}_{O \sim \nu}[\Gamma(OX)] = \frac{1}{n}J$, where $J$ is the all-ones matrix with $J_{ij} = 1(i, j = 1, 2, \cdots, n)$. Therefore, we have the expected variance as follows:

$$\begin{aligned}
\mathbb{E}_X[\mathrm{Var}_\Sigma(\hat{\beta}_\lambda \mid X)] &= \mathbb{E}_X\left[a(X;\lambda)^\top\frac{1}{n}Jb\right] \\
&= \frac{1}{n}\sum_{i,j=1}^n \mathbb{E}_X[a_i(X;\lambda)]b_j \\
&= \frac{1}{n}\mathbb{E}_X[\mathrm{Tr}((X^\top X + \lambda I)^{-1}X^\top X(X^\top X + \lambda I)^{-1}\Sigma)]\,\mathrm{Tr}(\Omega).
\end{aligned}$$

$\square$

Similarly, Theorem 3.5 can be extended as follows:

**Theorem E.2.** *Let Assumptions 2.1, and 3.2 hold. Then, we have*

$$\mathbb{E}_X[\mathrm{Var}(\hat{\beta}_\lambda \mid X)] = \frac{1}{np}\mathrm{Tr}(\Omega)\mathbb{E}_X[\mathrm{Tr}((\Lambda + \lambda I)^{-2}\Lambda)],$$

*where $XX^\top/p = U\Lambda U^\top$ for some orthogonal matrix $U \in O(n)$.*

