# OpenReview forum: "Prediction Risk and Estimation Risk of the Ridgeless Least Squares Estimator under General Assumptions on Regression Errors"
_ICLR.cc/2025/Conference — ICLR 2025 Poster_

### Official Review · Reviewer_CYv5 · 2024-10-25

**Soundness:** 3
**Presentation:** 3
**Contribution:** 2
**Rating:** 6
**Confidence:** 4

**Summary:**

This paper studies the prediction and estimation risks of the ridgeless least square regression under a more general assumption on the noise. Specifically, consider the data
$$
y_i = x_i^\top\beta + \epsilon_i,\ i=1,...,n
$$
with target $\beta\in\mathbb{R}^p$ and $ \epsilon_i$ is the noise independent of $x$ with $\mathbb{E}[\epsilon]=0$ and $\mathbb{E}[\epsilon\epsilon^\top]=\Omega$ finite and positive definite. This includes the case of i.i.d. Gaussian noise, autoregressive noise and cluster noise.

This paper applies the classical bias-variance decomposition onto the risks and finds a closed form expression for the variance term for both risks.

**Strengths:**

This paper provides rigorous proof and experiments to validate their claim. The assumption on the noise and the input is more general than previous works.

**Weaknesses:**

However, my biggest concern is the significance of the contribution provided by this paper.

As mentioned in line 132-133, this paper is not the first to consider non-i.i.d. Gaussian noise. Although this paper requires less assumptions on the noise than [1], it does not contain enough illustrations or interpretations on their main results (Theorem 3.4, 3.5) for "allowing potentially adversarial errors" as promised in line 142-143. Indeed, this paper does not explain how their main results could recover previous result with i.i.d. Gaussian noise or gain new insights with more general noise.

Also, it seems that the techniques used in the main results are rather standard and can be extended easily to more general settings like kernel ridge regression or kernel gradient flow, which could potentially increase the significance of this work.


Reference:
[1] Geoffrey Chinot and Matthieu Lerasle. On the robustness of the minimum ℓ2 interpolator.
Bernoulli, 2023.

**Questions:**

I believe this paper could improve its significance if it can answer the following questions:

1. What new insights could one gain from Theorem 3.4, 3.5?

2. Related to Q1, the true noise covariance $\Omega$ is not observable in reality. Could the authors provide any examples or algorithms to approximate such noise covariance in real-world datasets? If it is impractical to approximate it, how could we still analyse the risk with your risk expression in $\Omega$?

3. Could the authors extend their results to more general settings like kernel ridge regression or kernel gradient flow? Or at least explain what are the technical difficulties that might hinder the extension?

4. One Central idea of regularization is to balance out the effect of the noise in the labels. With the ridgeless linear regression setting, this paper unfortunately misses the opportunity to discuss the important interplay between the non-i.i.d. noise and regularization, which I find quite disappointing.

---

> ### Author Response · Authors · 2024-11-18
> **Answer 1/n**
>
> We thank the reviewer for the valuable feedback and insightful comments, but there is a **big misunderstanding**.
>
> > [W1] ... my biggest concern is the significance of the contribution provided by this paper. As mentioned in line 132-133, this paper is not the first to consider non-i.i.d. Gaussian noise. Although this paper requires less assumptions on the noise than [1], it does not contain enough illustrations or interpretations on their main results (Theorem 3.4, 3.5) for "allowing potentially adversarial errors" as promised in line 142-143. Indeed, this paper does not explain how  main results could recover previous result with i.i.d. Gaussian noise or gain new insights with more general noise.
> - [W1-A1] Misunderstanding
>     - First of all, we want to clarify that our noise is **NOT necessarily Gaussian**. For our main Theorem (Theorem 3.4 and 3.5), the only assumption we need is Assumption 2.1: $\mathbb{E}[\varepsilon]=0$ and $\Omega=\mathbb{E}[\varepsilon\varepsilon^\top]$ is finite and positive definite.
>     - We need neither Gaussianity nor i.i.d. assumption.
>     - Second, “allowing potentially adversarial errors” is **NOT our promise, but [1]'s**. [1] allows for potentially adversarial errors by not making any assumptions on the noise. Our motivation is **to accommodate time series and clustered data**, which are common in applications. We both do not make any assumptions on the noise, but with different motivations. “Allowing potentially adversarial errors” is not our goal. We take the corresponding sentences from the main text (line 141-145):
> > The main motivation of not making any assumptions on $\varepsilon_i$ in Chinot et al. (2022) and Chinot \& Lerasle (2023) is to allow for potentially adversarial errors. We aim to allow for a general variance-covariance matrix of the regression errors to accommodate time series  and clustered data, which are common in applications.
> - [W1-A2] Interpretations and New Insights
>     - So we answer with interpretations/insights of our main theorem for “allowing general noise”, taking the corresponding sentences from the main text (line 151-154):
> > Specifically, the variance term can be factorized into a product between two terms: one term depends only on the *trace* of the variance-covariance matrix, say $\Omega$, of $\varepsilon_i$'s; the other term is solely determined by the distribution of $x_i$'s.
> Interestingly, we find that **although $\Omega$ may contain non-zero off-diagonal elements, only the trace of $\Omega$ matters** (...).
> - [W1-A3] Recovering previous results
>     - It is easy to recover previous results. For example, **a special case of our result recovers the one in Hastie et al. (2022)**.
>     - If we restrict our general result to the case that the noise $\varepsilon_i$’s are i.i.d. (which implies that $\varepsilon$’s have the same variance and they are not correlated with each other) and $\Sigma=I$, then $\Omega\equiv \mathbb{E}[\varepsilon\varepsilon^\top]$ becomes a simple diagonal matrix $\sigma^2 I$. Then, in our results, $\text{Tr}(\Omega)$ recovers $n\sigma^2$ which leads to the same result in Lemma 1 of Hastie et al. (2022).
>     - We will upload a revised manuscript with the above discussion in a few days.
>
> $$\text{Var}_{\sigma^2 I}(\hat\beta\mid X)={\color{red}\sigma^2}\text{Tr}((X^\top X)^\dagger\Sigma)\quad \\text{Lemma 1 of Hastie et al. (2022)}$$
>
> $$\mathbb{E}\_X[\text{Var}_{\Sigma}(\hat\beta\mid X)]={\color{red}\frac{\text{Tr}(\Omega)}{n}}\mathbb{E}\_X[\text{Tr}((X^\top X)^\dagger\Sigma)]\quad \\text{Ours}$$
>
> > [W2] Also, it seems that the techniques used in the main results are rather standard and can be extended easily to more general settings like kernel ridge regression or kernel gradient flow, which could potentially increase the significance of this work.
> - [W2-A] Our main focus is the general assumption on the regression errors, so the suggestion on kernel regression or random features model (nonlinear transformation on the feature) is an **orthogonal research direction**.
> - See [Q4-A1] for the ridge regression setting.

---

> ### Author Response · Authors · 2024-11-18
> **Answer 2/n**
>
> > [Q1] What new insights could one gain from Theorem 3.4, 3.5?
> - [Q1-A] See W1-A2
>
> > [Q2] Related to Q1, the true noise covariance Ω is not observable in reality. Could the authors provide any examples or algorithms to approximate such noise covariance in real-world datasets? If it is impractical to approximate it, how could we still analyse the risk with your risk expression in Ω?
> - [Q2-A]
> Our goal is to understand the benefit of overparameterization under more general regression errors, which enables us **to accommodate more general data including time series, panel and grouped data** (see Figure 1, Example 2.1 and 2.2).
> To do so, we need to qualitatively analyze the risks, but it is not necessary to quantitatively approximate or estimate the $\Omega$. Actually, if the approximation is feasible then we can transform the noise and make it i.i.d. by using the inverse of $\Omega$.
> For example, from Theorem 3.4 and 3.5, we can conclude that the variance can be summarized through the trace of the $\Omega$, and the correlation between $\varepsilon_i$ and $\varepsilon_j$ (the off-diagonal term of $\Omega$) does not play any role as commented in W1-A2:
>  “the variance can be factorized into a product between two terms: one term depends only on the trace of the variance-covariance $\Omega$ of $\varepsilon_i$’s; the other term is solely determined by the distribution of $x_i$’s. Interestingly, we find that although $\Omega$ may contain non-zero off-diagonal elements, only the trace of $\Omega$ matters”
> To summarize, without the exact or estimated value of $\Omega$, we can provide a qualitative understanding that **the correlation $\mathbb{E}[\varepsilon_i\varepsilon_j]$ does not play any role**.
>
> > [Q3] Could the authors extend their results to more general settings like kernel ridge regression or kernel gradient flow? Or at least explain what are the technical difficulties that might hinder the extension?
> - [Q3-A] See W2-A1.
>
> > [Q4] One Central idea of regularization is to balance out the effect of the noise in the labels. With the ridgeless linear regression setting, this paper unfortunately misses the opportunity to discuss the important interplay between the non-i.i.d. noise and regularization, which I find quite disappointing.
> - [Q4-A1] Thank you for the suggestion. We obtain **similar results for the ridge estimator $\hat\beta_\lambda$** detailed as below:
>     $$\mathbb{E}\_X[\text{Var}\_\Sigma({\color{red} \hat\beta_\lambda}\mid X)]=\frac{1}{n}\text{Tr}(\Omega)\mathbb{E}\_X[\text{Tr}((X^\top X+\lambda I)^{-1}X^\top X(X^\top X+\lambda I)^{-1}\Sigma)]$$
> and
>     $$\mathbb{E}\_X[\text{Var}({\color{red} \hat\beta_\lambda}\mid X)]=\frac{1}{np}\text{Tr}(\Omega)\mathbb{E}\_X[\text{Tr}((\Lambda+\lambda I)^{-2}\Lambda)].$$
> Note that the limit $\lambda\rightarrow 0$ recovers our original results (cf. $A^\dagger A A^\dagger = A^\dagger$).
> - We will add the details with the proofs in the revised paper (see Appendix E). The corresponding asymptotic result recovers Corollary 6 (but with $\lim_{p\rightarrow\infty}\text{Tr}(\Omega)/p$ instead of $\sigma^2$) in Hastie et al. 2022 for the isotropic features.
> - [Q4-A2] It is important to consider the ridgeless setting **to understand the generalization ability of overparameterized model** (e.g., deep neural networks) **trained with gradient-based optimization** (e.g., gradient descent), also known as “double descent “ or “benign overfitting”.
> - We consider the ridgeless setting because the minimum $\ell_2$ norm least squares estimator $\hat\beta$ under the ridgeless setting is **directly connected to gradient descent** as described below:
>     - **The limit $\lim_{k\rightarrow\infty}\beta^{(k)}$ obtained by running gradient descent** with a proper learning rate on the least squares loss starting from $\beta^{(0)}=0$ is **the minimum norm solution**.
>     - In other words, gradient descent has an implicit bias toward the minimum norm solution.
>     - We usually use a small initialization $\beta^{0}$. Then we can obtain a solution close to the minimum norm solution.
>     - It is important to note that without the regularization effect, we can obtain small prediction/estimation risks from the overparameterization. This is called **implicit regularization**.
> - In a similar sense, we quote some sentences from Hastie et al. (2022) and Gunasekar et al. (2017).
> > “While in low dimension a positive regularization is needed to achieve good interpolation, in certain high-dimensional settings interpolation can be nearly optimal.“
>
>     > "(...) using gradient descent to optimize an unregularized, underdetermined least squares problem would yield the minimum Euclidean norm solution (...)"
> - We will clarify this point and add the above discussion in the revised manuscript.
>
> **[Reference]**
> - Gunasekar et al. (2017), Implicit regularization in matrix factorization (NeurIPS 2017)

---

> ### Comment · Reviewer_CYv5 · 2024-11-20
>
> Sorry for my misunderstanding on the paper and thank you for your detailed answer. I will raise my score accordingly.

---

> > ### Author Response · Authors · 2024-11-22
> >
> > Thank you for acknowledging our response and raising the score. We are pleased that we have addressed your concerns.

---

### Official Review · Reviewer_Wss5 · 2024-10-30

**Soundness:** 4
**Presentation:** 4
**Contribution:** 3
**Rating:** 8
**Confidence:** 3

**Summary:**

The paper extends the analysis of ridgeless least squares estimators to more
realistic error structures beyond the traditional i.i.d. assumptions. It
addresses both prediction and estimation risks in settings where regression
errors may be correlated or exhibit heteroscedasticity.

The authors provide exact finite-sample expressions for both types of risk,
which are decomposed into bias and variance components. The variance in
prediction risk is shown to be the product of two distinct terms: one related to
the error covariance matrix and the other dependent on the feature distribution.

The paper also conducts a systematic asymptotic analysis of prediction and
estimation risks. Numerical experiments with autoregressive and clustered
regression errors illustrate the theoretical findings.

**Strengths:**

Originality:

This paper extends the analysis of ridgeless least squares estimators by
relaxing the i.i.d. assumption on the regression errors. Correlated and
heteroscedastic errors are frequently encountered in practice but are not
investigated in prior works in the high-dimensional setting. This work fills a
gap in understanding the performance of least squares estimators, extending the
observation of double descent to more general settings.

Quality:

The paper is technically sound and comprehensive in its treatment of both
prediction and estimation risks. The finite-sample and asymptotic behaviors
finite-sample results are rigorously presented. The numerical result in Section
1 demonstrates the motivation of the generalized assumptions and the results in
Section 3.3 help validate the theoretical findings.

Clarity:

The paper is overall well written with clear presentation.

Significance

The paper extends the theory of least squares estimators to include the
situation with non i.i.d. errors in the context of overparameterization. It fill
a gap in the literature, so the results deserve to be documented.

**Weaknesses:**

For the high-dimensional setting considered in the paper, ridgeless least
squares estimators are seldom applied in practice. Regularization is almost
always helpful, and there is a large body of literature demonstrating the
advantages of regularized least squares. The paper should at least discuss this
line of research. In particular, it has been shown that optimized ridge
regression avoids bias inflation. Why, then, would practitioners care about
ridgeless least squares estimators in this scenario?

While the assumptions are more general than the i.i.d. case, the scenario
considered is still much simpler than what is encountered with real, practical
data. For example, the assumption of left-spherical symmetry for the design
matrix is limiting in practical scenarios. Most real-world datasets feature
asymmetrical or skewed distributions, and many variables in real data are
actually categorical, rather than numerical. It would help to add discussions on
the limitations of the current investigation in these contexts.

Most of the theoretical results, especially the finite-sample results, appear
similar to the low-dimensional or fixed-$p$ case. It would be helpful to include
discussions on this point. If the results are not the same, do they reduce to
the low-dimensional results? Even if they have the same expression, it would be
insightful to explain why it is nontrivial to derive these results in the
high-dimensional setting.

**Questions:**

1. Include discussions on relevant regularized least squares estimators, and
   provide scenarios where the ridgeless least squares estimator excels or
   underperforms.
2. Provide comparative experiments with other regularization techniques to
   demonstrate where ridgeless least squares excels or underperforms.
3. Add discussions on the limitations of the current investigation, particularly
   in terms of the assumptions.
4. Provide discussions connecting the results in the high-dimensional setting to
   those in the low-dimensional setting.

---

> ### Author Response · Authors · 2024-11-18
> **Answer 1/n**
>
> We thank the reviewer for the valuable feedback and insightful comments.
>
> > [W1] For the high-dimensional setting considered in the paper, ridgeless least squares estimators are seldom applied in practice. Regularization is almost always helpful, and there is a large body of literature demonstrating the advantages of regularized least squares. The paper should at least discuss this line of research. In particular, it has been shown that optimized ridge regression avoids bias inflation. Why, then, would practitioners care about ridgeless least squares estimators in this scenario?
> - [W1-A1] Thank you for the suggestion. We obtain **similar results for the ridge estimator $\hat\beta_\lambda$** detailed as below:
> $$\mathbb{E}\_X[\text{Var}\_\Sigma({\color{red}\hat\beta_\lambda}\mid X)]=\frac{1}{n} \text{Tr}(\Omega)\mathbb{E}\_X[\text{Tr}((X^\top X+\lambda I)^{-1}X^\top X(X^\top X+\lambda I)^{-1}\Sigma)]$$
> and
>     $$\mathbb{E}\_X[\text{Var}({\color{red}\hat\beta_\lambda}\mid X)]=\frac{1}{np} \text{Tr}(\Omega)\mathbb{E}_X[\text{Tr}((\Lambda+\lambda I)^{-2}\Lambda)].$$
> The limit $\lambda\rightarrow 0$ recovers our original results (cf. $A^\dagger AA^\dagger =A^\dagger$).
>
> - We will add the details with the proofs in the revised paper (see Appendix E).
> - [W1-A2] It is important (also for practitioners) to consider the ridgeless setting **to understand the generalization ability of overparameterized model** (e.g., deep neural networks) **trained with gradient-based optimization** (e.g., gradient descent), also known as “double descent“ or “benign overfitting”.
> - We consider the ridgeless setting because the minimum $\ell_2$ norm least squares estimator $\hat\beta$ under the ridgeless setting is **directly connected to gradient descent** as described below:
>     - **The limit $\lim_{k\rightarrow\infty}\beta^{(k)}$ obtained by running gradient descent** with a proper learning rate on the least squares loss starting from $\beta^{(0)}=0$ is **the minimum norm solution**.
>     - In other words, gradient descent has an implicit bias toward the minimum norm solution.
>     - We usually use a small initialization $\beta^{0}$. Then we can obtain a solution close to the minimum norm solution.
>     - It is important to note that without the regularization effect, we can obtain small prediction/estimation risks from the overparameterization. This is called **implicit regularization**.
>
> - In a similar sense, we quote some sentences from Hastie et al. (2022) and Gunasekar et al. (2017).
> > “While in low dimension a positive regularization is needed to achieve good interpolation, in certain high-dimensional settings interpolation can be nearly optimal.“
>
>     > "(...) using gradient descent to optimize an unregularized, underdetermined least squares problem would yield the minimum Euclidean norm solution (...)"
> - We will clarify this point and add the above discussion in the revised manuscript.
>
>
> > [W2] While the assumptions are more general than the i.i.d. case, the scenario considered is still much simpler than what is encountered with real, practical data. For example, the assumption of left-spherical symmetry for the design matrix is limiting in practical scenarios. Most real-world datasets feature asymmetrical or skewed distributions, and many variables in real data are actually categorical, rather than numerical. It would help to add discussions on the limitations of the current investigation in these contexts.
> - [W2-A] There is a misunderstanding. The left-spherical symmetry assumption does **not exclude asymmetrical or skewed distributions**. The (feature-dimension-wise) asymmetry or skewness of each $x_i$ is a different concept with (sample-wise) left-spherical symmetry of $X$.
> - Thank you for the suggestion. We will add discussions on the limitations of the current investigation.
>
> > [W3] Most of the theoretical results, especially the finite-sample results, appear similar to the low-dimensional or fixed-p case. It would be helpful to include discussions on this point. If the results are not the same, do they reduce to the low-dimensional results? Even if they have the same expression, it would be insightful to explain why it is nontrivial to derive these results in the high-dimensional setting.
> - [W3-A] Under the low-dimensional setting (underparameterized case, $p<n$),
> the estimator is $\hat\beta = (X^\top X)^{-1} X^\top y = \beta + (X^\top X)^{-1} X^\top\varepsilon$ where $y = X\beta +\varepsilon$
> The bias $\text{Bias}\_\Sigma(\hat\beta\mid X)=\\|\mathbb{E}[\hat\beta\mid X]-\beta\\|_\Sigma$ is zero (as $X^\top X$ is invertible, i.e., $\mathbb{E}[\hat\beta\mid X]=\beta$).
> The risk is just variance. Therefore, the results in Theorem 3.4, 3.5 are the same, but now we can write **the inverse matrices** $(X^\top X)^{-1}$ and $\Lambda^{-1}$ instead of **the pseudoinverse matrices** $(X^\top X)^\dagger$ and $\Lambda^\dagger$.
>
> **[Reference]**
> - Gunasekar et al. (2017), Implicit regularization in matrix factorization (NeurIPS 2017)

---

> > ### Comment · Reviewer_Wss5 · 2024-11-26
> >
> > Thank you for the clarification.

---

> ### Author Response · Authors · 2024-11-18
> **Answer 2/n**
>
> > [Q1] Include discussions on relevant regularized least squares estimators, and provide scenarios where the ridgeless least squares estimator excels or underperforms.
> - [Q1-A] See [W1-A1/A2]
>
> >[Q2] Provide comparative experiments with other regularization techniques to demonstrate where ridgeless least squares excels or underperforms.
> - [Q2-A] We would like to clarify the focus and scope of our paper. Specifically, our work does not propose a novel estimation method but instead offers a new and rigorous analysis of the widely-used ridgeless least squares estimator. The primary contribution lies in the theoretical examination of this popular estimator under a more general and realistic assumption about the regression error structure. Given this focus, our goal is not to compare the ridgeless least squares estimator with other existing regularization methods, but rather to deepen the understanding of its performance in scenarios that reflect broader practical applicability.
>
> > [Q3] Add discussions on the limitations of the current investigation, particularly in terms of the assumptions.
> - [Q3-A] See [W2-A]
>
> > [Q4] Provide discussions connecting the results in the high-dimensional setting to those in the low-dimensional setting.
> - [Q4-A] See [W3-A]

---

### Official Review · Reviewer_4djy · 2024-11-02

**Soundness:** 3
**Presentation:** 3
**Contribution:** 3
**Rating:** 6
**Confidence:** 4

**Summary:**

In this paper, the authors investigate prediction risk and  estimation risk under more general regression error assumptions beyond i.i.d. errors. In particlar, they explore the benefits of overparameterization in a more realistic setting, which allows for clustered or serial dependence. This paper  demonstrated   that the estimation difficulties associated with the variance components  can be summarized through the trace of the variance-covariance matrix of the regression errors.

**Strengths:**

The paper is written clearly.  I appreciate the significant contribution to high-dimensial data analysis.  The new approach is  promising for
 more  broad framework. This paper attacked the very challenging problem involved in least-squares estimators beyond the  assumiption with  i.i.d.  errors. The new idea about  benefits of  over-parameterization could be extended  to time series, panel and grouped data, etc.  with the broad impact.

**Weaknesses:**

The paper proved several  good theoretical results and properties. The experiemts of data set may not be comprehensive, Since there is no comparison of  the  proposed method with existing methods.   In addition,  the proposed approach only works for  the case  p>n. In practice,    the ultra-high dimensional case with p=exp(n) is very  common.  This is a more realistic setting in the big data era.

**Questions:**

I have several comments and suggestions for the authors to address.


1. The paper is not complete. Please add the conclusion section in the revision.

2. It is worthwhile to extend the proposed approach from the case  p>n   to the ultra-high dimensional case with p=exp(n).

3. It is of interest for the authors to  compare   proposed method with existing ones  (including  Chinot et al. (2022) and Chinot & Lerasle (2023), etc.)  in the experiments.

4. The new idea about benefits of over-parameterization could be extended to time series, panel and grouped data.  It is helpful  for the authors to  elaborate it by proving more details.

5. There are some  typos, grammatical errors,  etc. in the paper. Please check it in the revision carefully.

in page 1, line 022, "can extend" -> "can be extended".

in page 5, line 312, "appendix" -> "Appendix".

in page 9, line 432, "appendix" -> "Appendix".

in page 10, line 500, "appendix" -> "Appendix".

in page 14, line 704, "Figure" -> "Figures".

in page 17, line 905, "6" -> "(6)".

---

> ### Author Response · Authors · 2024-11-18
> **Answer 1/n**
>
> We thank the reviewer for the valuable feedback and insightful comments.
>
> > [W1] The paper proved several good theoretical results and properties. The experiemts of data set may not be comprehensive, Since there is no comparison of the proposed method with existing methods.
>
> - [W1-A] Thank you for raising this issue. We would like to clarify the focus and scope of our paper. Specifically, our work does not propose a novel estimation method but instead offers a new and rigorous analysis of the widely-used ridgeless least squares estimator. The primary contribution lies in the theoretical examination of this popular estimator under a more general and realistic assumption about the regression error structure. Given this focus, our goal is not to compare the ridgeless least squares estimator with other existing methods, but rather to deepen the understanding of its performance in scenarios that reflect broader practical applicability.
>
> > [W2] In addition, the proposed approach only works for the case p>n. In practice, the ultra-high dimensional case with p=exp(n) is very common. This is a more realistic setting in the big data era.
>
> - [W2-A] Thank you for your comment regarding the ultra-high dimensional case with $p = \exp(n)$. We believe this observation may stem from a misunderstanding of the scope and results of our paper.
>
> - First, our key results (e.g., Theorems 3.4 and 3.5) pertain to a **finite-sample analysis**, where $p$ and $n$ are fixed. This inherently accommodates the case of $p = \exp(n)$. The only results constrained by asymptotic assumptions are in Section 4.2.1, where we consider the regime $p/n \to \gamma > 1$.
>
> - Additionally, we would like to highlight that the ultra-high dimensional case of $p = \exp(n)$ is commonly studied under a sparsity assumption, where the number of non-zero $\beta$ coefficients is much smaller than the sample size $n$, or similar structural regularity conditions. For example, this is discussed extensively in Bühlmann and van de Geer (2011), *Statistics for High-Dimensional Data: Methods, Theory, and Applications* (Springer).
>
> - In summary, our theoretical analysis is not constrained as suggested:
>
> 1. Our main results in the **non-asymptotic setting** require only that $p > n$, and naturally accommodate $p = \exp(n)$.
> 2. Our **asymptotic analysis** in Section 4.2.1, while considering $p/n \to \gamma > 1$, does not rely on sparsity or similar regularity conditions, thus offering complementary insights.
>
> - We hope this clarification resolves any misunderstanding about the scope and generality of our results.
>
> > [Q1] The paper is not complete. Please add the conclusion section in the revision.
>
> - [Q1-A] We will add the conclusion part in the revised manuscript in a few days.
>
> > [Q2] It is worthwhile to extend the proposed approach from the case p>n to the ultra-high dimensional case with p=exp(n)
>
> - [Q2-A] See [W2-A].
>
> > [Q3] It is of interest for the authors to compare proposed method with existing ones (including Chinot et al. (2022) and Chinot & Lerasle (2023), etc.) in the experiments.
>
> - [Q3-A] See [W1-A].
>
> > [Q4] The new idea about benefits of over-parameterization could be extended to time series, panel and grouped data. It is helpful for the authors to elaborate it by proving more details.
>
> - [Q4-A] Thank you for your thoughtful suggestion. As it is unclear whether "proving more details" was intended to mean "providing more details," we have interpreted the comment as a request for additional elaboration.
> - In fact, our paper already includes examples designed to provide detailed insights into these cases. Specifically:
>     - Example 2.1 (Time Series - AR(1) Errors) on page 4 offers a simple yet illustrative example for time series data.
>     - Example 2.2 (Panel and Grouped Data - Clustered Errors) on page 4 addresses scenarios involving both panel and grouped data.
> - To make this connection clearer, we will revise the paper to explicitly state that Example 2.1 pertains to time series data, while Example 2.2 is relevant to panel and grouped data. We hope this clarification satisfies the reviewer's concern and highlights the applicability of our ideas to these contexts.
>
> > [Q5] There are some typos, grammatical errors, etc. in the paper. Please check it in the revision carefully. (...)
>
> - Thank you for the comments. We will revise the paper accordingly.

---

> ### Comment · Reviewer_4djy · 2024-11-26
>
> Thank you very much  for clarifying.  I would like to retain my score.

---

### Author Response · Authors · 2024-11-20
**Global Comments**

We would like to thank the reviewers for the thorough examination of the paper and their insightful and valuable comments.

We appreciate that the reviewers recognized the strengths of our paper, saying the paper is "well-written with clear presentation", "sound and comprehensive", it "attacked the very challenging problem", and "good theoretical results and properties" are "rigorously presented".
Moreover, "the numerical results (...) demonstrate the motivation" and "help validate the theoretical findings".
They also said our paper is "new" and "promising", and it "extends the theory (...) to time series, panel and grouped data, etc. with the broad impact (...) beyond the assumption with i.i.d. errors". They also recognize the novelty, saying that our setting on the regression errors is "more general than previous works" and "frequently encountered in practice but are not investigated in prior works in the high-dimensional setting", so this paper "fills a gap in the literature".

During the author response period, we have given careful thought to the reviewers’ suggestions to answer the questions and concerns (we will make the corresponding revisions to our manuscript):
- We clarify our focus and restate/emphasize our contributions.
- We fix some typos.
- We add Conclusion (Section 5) and additional results and proof for ridge regression setting (Appendix E).

---

### Meta-Review · Area_Chair_cgDn · 2024-12-22

**Metareview:**

This paper studies both the prediction risk as well as the estimation risk in non-trivial minimum l2 norm or 'ridgeless' least squares problems. In particular, they consider a general setting that leads to a better understanding of non-iid settings, including a relationship between estimating variance components and a trace representation. This treatment applies to several types of grouped data, as well as for error structures arising from longitudinal correlation. Reviewers are largely positive and I agree with their assessment of the strengths in this article.

**Additional Comments On Reviewer Discussion:**

Reviewers engaged briefly in the rebuttal phase

---

### Decision · Program_Chairs · 2025-01-22

Accept (Poster)